# Transcatheter Edge-to-Edge Repair of the Mitral Valve in Four Dogs: Preliminary Results Regarding Efficacy and Safety

**DOI:** 10.3390/ani14213068

**Published:** 2024-10-24

**Authors:** Soontaree Petchdee, Wanpitak Pongkan, Jing Lei, Kotchapol Jaturanratsamee, Ratikorn Bootcha, Wannisa Meepoo, Chattida Panprom

**Affiliations:** 1Department of Large Animal and Wildlife Clinical Science, Faculty of Veterinary Medicine, Kasetsart University, Kamphaeng Saen, Nakorn Pathom 73140, Thailand; 2Faculty of Veterinary Medicine, Chiang Mai University, Chiang Mai 50200, Thailand; 3Shanghai Xinyu Pet Hospital, Shanghai 201109, China; 4Science and Innovation for Animal Health Program, Faculty of Veterinary Medicine, The Graduate School, Kasetsart University, Kamphaeng Saen, Nakorn Pathom 73140, Thailand; 5Kasetsart University Veterinary Teaching Hospital, Faculty of Veterinary Medicine, Kasetsart University, Kamphaeng Saen, Nakorn Pathom 73140, Thailand

**Keywords:** dog, heart, mitral valve, surgical intervention, v-clamp

## Abstract

Mitral valve regurgitation is a common prevalent heart condition in dogs that often progresses to heart failure. Although medical treatment is the recommended therapy, and, in the past few years, open heart surgery has been one therapeutic option for repairing the mitral valve, but the procedure is invasive and time-consuming. Recently, a new mitral valve repair device has been introduced. This technique can immediately repair the mitral valve and might be used as another treatment option to reduce the incidence of complications of open-heart surgery in dogs. Although additional safety information is needed, the results suggest that the procedure is safe, potentially eliminating the need for heart failure medication after repair.

## 1. Introduction

As one of the most common heart diseases in dogs, myxomatous mitral valve disease (MMVD) is associated with symptoms such as cough and increased respiratory effort. In dogs, MMVD causes progressive degeneration of the mitral valve. The mitral valve is composed of two leaflets, the anterior and posterior leaflets. Both leaflets are divided into three zones: A1, A2, and A3, and P1, P2, and P3. Abnormal characteristics of the mitral valve leaflet include elongated chordae tendinae, thickening, and shrinkage of the leaflets with bulging or prolapse toward the left atrium [1]. Medication is the standard treatment for this disease, which delays the presentation of heart failure and the heart remodeling process. Therefore, common treatment practices and ACVIM guidelines recommend individualized medication treatment [2,3].

In recent years, open heart surgical treatment has been performed for mitral valve repair in dogs; some recommend surgery for patients with B2 disease [4]. However, similar to surgery in humans, open heart surgery in dogs is invasive, time-consuming, and costly to owners without pet insurance. The surgical procedure requires heart and lung bypass and carries a high risk of complications such as bleeding and thrombosis [5]. Percutaneous mitral valve repair via the MitraClip^®^ system (Abbott Vascular, Abbott Park, Chicago, IL, USA) was established in 2008. The use of the MitraClip for congestive heart failure (CHF) as a result of mitral valve dysfunction treatment has been shown to be safe and less invasive, relieve clinical symptoms, and reduce the length of hospital admission [6,7]. Unfortunately, MitraClip is not available for use in dogs because of the size of the device. A transcatheter edge-to-edge mitral valve repair (TEER) device called the ValveClamp was designed to treat structural mitral valve disease in humans and has been used in the clinical setting in China [8]. A TEER device named the V-clamp was established for clinical use in dogs by HongYu Medical Technology, Shanghai, China. The V-clamp is designed for canine mitral valve regurgitation and is most appropriate when regurgitation is limited to a specific leaflet, particularly when there is no leaflet cleft. The V-clamp procedure is considered to have a lower risk than open heart surgery because it avoids the use of cardiopulmonary bypass to stop the heartbeat and requires less surgical time.

The selection criteria for the V-clamp procedure that have the highest likelihood of a successful outcome include small breed dogs with ACVIM stage B2 or stage C weighing over 3 kg, or the diameter of the anterior-posterior (AP) mitral valve leaflet measured via transthoracic echocardiography (TTE) greater than 10 mm.

Minnesotans Living with Heart Failure and Validation has been extensively used, demonstrating strong reliability and validity in assessing health-related quality of life in chronic heart failure patients in humans [9]. The Minnesotans Living with Heart Failure and Validation items are scored by adding the individual item scores for each dimension, resulting in a total score as well as separate scores for physical and emotional well-being. A higher total score indicates a lower overall quality of life. The evaluation results from the questionnaires, which were adapted from Minnesotans Living with Heart Failure and Validation, may help with further consideration of postoperative quality of life as an important goal and might be used as an adjunct to quantitative outcomes for dogs.

## 2. Materials and Methods

### 2.1. Physical Examination and Laboratory Investigations

All dogs were assessed for a physical examination (inspection, palpation, percussion, and auscultation) for general health status. In this evaluation, mucous membranes, capillary refill time, rectal temperature, heart rate, heart sounds, pulse rate, and state of hydration were assessed. The dogs presented with a history of coughing, dyspnea, and exercise intolerance. Thoracic radiography revealed left heart enlargement (Figure 1). The mitral valve structure was evaluated by 3D transesophageal echocardiography, and a satisfactory clamp position was confirmed using fluoroscopy (GE, Boston, MA, USA). and 3D-TEE (Philips Medical Systems, Andover, MA, USA) as shown in Figure 2 and Figure 3.

Blood samples were collected by venipuncture and submitted for routine hematological and biochemical investigations. All blood samples were then separated into two fractions. The first fraction was put into tubes containing potassium ethylene diamine tetra-acetic acid (EDTA) to evaluate the complete blood count (CBC) analysis. The second fraction was put in tubes containing silica dioxide to evaluate blood chemistry analysis (blood urea nitrogen (BUN), creatinine (Cre), alanine aminotransferase (ALT), alkaline phosphatase (ALP), total protein (TP), albumin (ALB), and globulin (GLB)). This study was conducted at the Faculty of Veterinary Medicine, Kasetsart University Kamphaengsaen. Written client consent to participate was obtained from each patient owner enrolled in this surgical treatment.

### 2.2. Thoracic Radiography

All radiographs were obtained using a digital radiography system, with dogs positioned in right lateral recumbency, and dorsoventral views. Thoracic radiographic images were captured using a GE Revolution XR/d digital X-ray system, (GE, Boston, MA, USA). operated at tube potentials from 60 to 130 kVp. The lung field, dilation of the pulmonary artery and vein, elevation of the distal part of the trachea toward the spine, and the presence of cardiomegaly were assessed. The size of the heart and left atrium were determined using the vertebral heart scale (VHS) and vertebral left atrium size (VLAS). Thoracic radiography was performed to determine the heart position and to locate the cardiac apex, to ensure precise incision placement for minimally invasive surgery and thereby reducing procedure time. Thoracic radiography images were assessed both before and after surgery to evaluate radiographic features, such as alterations in the vascular pattern and pulmonary atelectasis, a common complication in patients undergoing heart surgery [10] as illustrated in Figure 1.

### 2.3. Echocardiography

Pre- and post-operative, the patients underwent transthoracic echocardiography (TTE) by using a General Electric Vivid 5s ultrasound system (GE, Boston, MA, USA)with continuous electrocardiography monitoring to evaluate the cardiac function. A probe size of 6 MHz was used in this study. Echocardiography was performed at baseline, and every month until 5–6 months after the surgical intervention by two skilled sonographers. The measurement was performed in a right parasternal long axis and right parasternal short axis in a right lateral recumbent position, whereas and left apical four-chamber view was performed in a left lateral recumbent position without sedation. Echocardiographic images were captured and stored for offline analysis.

In the right parasternal long and short-axis view, the left ventricular wall structure and function were calculated by measuring the images from two-dimensional and M-mode planes. LV wall thickness, LV dimension, and LV function were evaluated by M-mode echocardiography in the right parasternal short axis (at the base of the heart and the level of the papillary muscles) [11]. Ventricular wall thickness and dimensions were recorded during diastole and systole to obtain the parameters such as diastolic interventricular septum thickness (IVSd), systolic interventricular septum thickness (IVSs), left ventricular end-diastolic diameter (LVIDd), left ventricular end-systolic diameter (LVIDs), left ventricular wall diastolic thickness (LVPWd), and left ventricular wall systolic thickness (LVPWs). All averaged M-mode chamber measurements were normalized by body weight using the Cornell allometric scale for dogs [12]. In addition, M-mode echocardiography parameters were used to calculate the percentage of fractional shortening (%FS) and percentage ejection fraction (%EF) by using the Teicholz formula which was accomplished automatically by the echocardiographic equipment software (EchoPAC™). The right parasternal short-axis view was used to measure the left atrial dimension (LA) and aortic dimension (AO) in early diastole. Then the left atrial to aortic root ratio (LA: AO ratio) was calculated using the Swedish method. Three consecutive beats of cardiac cycles were measured, and the average values were used for all echocardiographic parameters. Right ventricular dimensions and pulmonary artery diameters were observed.

In the left apical four-chamber view, the diastolic function indicated by the E-wave per A-wave ratio was performed in the left apical four-chamber view using a pulse wave (PW) Doppler technique. Pulmonary velocity was evaluated using a PW Doppler technique. Moreover, we investigated and calculated the left ventricular outflow tract (LVOT) and mitral regurgitation (MR) rates to determine the LVOT stroke volume and the MR stroke volume before and after surgery in all cases.

Perioperative 3D-transesophageal echocardiogram (TEE) was performed using a Philips Epiq 7C ultrasound machine with a transducer from 1 to 7 MHz (Philips Medical Systems, Andover, MA, USA). TEE was obtained to plan the procedure and confirm the appearance of the heart valves, including the leak location.

### 2.4. Anesthetic and Analgesic Protocol

Anesthetics and analgesics play a crucial role in thoracic surgery for cardiac patients. Ensuring effective analgesia is also important for preventing hemodynamic changes caused by surgical pain [13]. Patients were premedicated with Butorphanol; Zoetis, 10 mg/mL (0.2 mg/kg) and then followed by etomidate; B BRAUN, 2 mg/mL (1–2 mg/kg) for induction and intercostal block with bupivacaine; Aspen, 5 mg/mL (at ribs 5, 6, 7, 8, and 9). The anesthesia was maintained by isoflurane at a 2–5% concentration in oxygen at 1.5–2 L/min. In our study, all patients underwent continuous rate infusion (CRI) of potent opioids (2–5 mcg/kg of Fentanyl; Hamein) along with intercostal nerve block. However, monitoring for potential side effects of potent opioids, such as bradycardia, is essential. Low-dose atropine might be used as an intervention if necessary.

### 2.5. Surgical Intervention

Surgical intervention was performed using a V-clamp device (HongYu Medical, Shanghai, China) in four patients. The surgical procedure has been described previously [14,15]. In brief, an incision was made at the left 5th intercostal space. A 2-0 polypropylene monofilament purse-string suture was placed at the puncture area and then a 14 French sheath introducer (HongYu Medical Technology, Shanghai, China) was guided into the left ventricle, and the V-clamp device was inserted through the sheath and positioned in the left ventricle under transesophageal echocardiography and 3D imaging (Figure 2). Limb-lead electrocardiography was used to determine the cardiac rhythm. The mitral valve leaflet was repaired by V-clamp. After the mitral valve was clipped, the guiding catheter was removed and flushed, and then the skin was sutured with a surgical nylon. A chest drainage tube and a urinary catheter were placed after completing the surgical procedure. Chest drainage is used for the removal of air, fluid, or blood. Post-operative urine monitoring is also important for identifying any post-operative complications. Patients were recovered in a cage filled with oxygen for 24–48 h. For 24 h after surgery, the chest drainage and the urinary catheter were removed. The blood profiles and thoracic radiographs were evaluated every day until discharge.

### 2.6. Statistical Analysis

Data were represented as mean ± standard deviation (SD). The X column represents time, and the response at each time point is entered in four patients for the Echocardiography parameters using GraphPad Prism 9 software.

## 3. Results

The owners evaluated questionnaires adapted from Minnesotans Living with Heart Failure and Validation [9] to assess various aspects of quality of life. These questionnaires measure happiness, mental status, hygiene, and physical fitness, with higher scores indicating a lower quality of life, as detailed in Table 1. The dogs exhibited improvements in respiratory signs during the follow-up visit and showed increased happiness and overall quality of life after the surgical intervention, as shown in Table 1.

Prior to repairing the mitral valve, a biochemical blood profile revealed elevated liver enzymes in all dogs. However, the profiles, particularly the liver enzymes, improved one month after the valve was repaired, as shown in Table 2.

The medications, including diuretics and pimobendan, were discontinued, and only clopidogrel was prescribed for all patients at the time of discharge. However, one dog developed right-sided congestive heart failure due to previous tricuspid regurgitation, necessitating the represcription of diuretics and pimobendan for that patient for 60 days until the clinical symptoms such as ascites were improved.

**Patient 1:** A nine-year-old female Beagle dog with a previous history of cough and evidence of ACVIM stage B2 in May 2023. Before mitral valve repair, her prescription included 0.25 mg/kg pimobendan every 12 h, 1.0 mg/kg spironolactone daily, 2.0 mg/kg furosemide daily, and cardiac support, including coenzymes Q10 and L-carnitine. Transthoracic echocardiography revealed stage B2 disease characterized by LA dilation and mild mitral valve regurgitation. Three-dimensional transesophageal echocardiography was used to observe the main regurgitation orifice. Under general anesthesia, V-clamp type IV was applied to the center of the anterior leaflet (A2) and the center of the posterior leaflet (P2) to reduce regurgitation. After mitral valve repair, the mean transmitral gradient was 1 mmHg. The dog was discharged twenty-four hours after the operation. At the one-month follow-up examination, the dog exhibited remarkable clinical improvement. The size of the left atrium decreased by 13.7%, 17.6%, and 5.9% at one, two, and three months after mitral valve repair, respectively.

**Patient 2:** A fourteen-year-old female Pomeranian dog with a previous history of cough. Before mitral valve repair, his prescription included 0.25 mg/kg pimobendan q12h, 1.0 mg/kg daily spironolactone, 2.0 mg/kg furosemide q12h, cardiac support including coenzymes Q10 and L-carnitine, and liver support. Transthoracic echocardiography revealed stage C disease, characterized by LA and LV dilation and severe mitral valve regurgitation. Three-dimensional transesophageal echocardiography was used to observe the main regurgitation orifice. The anterior leaflet of the mitral valve prolapsed with a 22.8 anteroposterior (AP) diameter. The length of the anterior leaflet was 13.2 mm, and the length of the posterior leaflet was 9.9 mm. V-clamp type IV was used to reduce regurgitation. The center of the anterior leaflet (A2) and the center of the posterior leaflet (P2) were deployed by the V-clamp device to reduce regurgitation under general anesthesia. After mitral valve repair, the mean transmitral gradient was 4 mmHg. A ventricular escape rhythm with a heart rate of 120 bpm was observed after mitral valve repair, and a 2 mg/kg bolus of lidocaine along with 2 mcg/kg/h fentanyl was intravenously administered to control pain and arrhythmias. The dog was discharged on post-operative day 5. At the one-month follow-up examination, the dog exhibited remarkable clinical improvement in the heart dimension, and the size of the LA decreased by 19.5%, 17.9%, and 6.7% at one, two, and three months after mitral valve repair, respectively.

**Patient 3:** A nine-year-old female Chihuahua with a previous history of cough and increased respiratory effort. Before mitral valve repair, her prescription included 0.25 mg/kg pimobendan every 12 h, 1.0 mg/kg spironolactone daily, 2.0 mg/kg furosemide every 12 h, and cardiac support, including coenzyme Q10 and L-carnitine. Transthoracic echocardiography revealed stage C disease, characterized by LA and LV dilation, severe mitral valve regurgitation, and moderate tricuspid valve regurgitation with increased pulmonary artery pressures. Three-dimensional transesophageal echocardiography was used to observe the main regurgitation orifice. No flail of the mitral valve was observed with a 21.7 AP diameter. The length of the anterior leaflet was 7.7 mm, and the length of the posterior leaflet was 5.8 mm. Under general anesthesia, V-clamp type IV was applied at the center of the anterior leaflet (A2), and the center of the posterior leaflet (P2) was deployed to reduce regurgitation. After mitral valve repair, the mean transmitral gradient was 1 mmHg. The dog was discharged 48 h after surgery. At the one-month follow-up examination, the dogs exhibited remarkable clinical improvement in the left ventricle dimension by 8.3%, 11.4%, and 25.4% at one, two, and three months after mitral valve repair, respectively. The patient exhibited a stable mitral valve V-clamping position, moderate regurgitation, and fractional shortening of 55.7%.

**Patient 4:** An eleven-year-old male Poodle dog with a previous history of cough and evidence of ACVIM stage B2 in April 2022. Before mitral valve repair, his prescription included 0.25 mg/kg pimobendan every 12 h, 1.0 mg/kg daily spironolactone, 2.0 mg/kg daily furosemide, and cardiac support, including coenzymes Q10 and L-carnitine. Transthoracic echocardiography revealed stage B2 disease, characterized by LA dilation and moderate mitral valve regurgitation. Three-dimensional transesophageal echocardiography was used to observe the main regurgitation orifice. The anterior leaflet of the mitral valve prolapsed, with a 12.7 AP diameter. The length of the anterior leaflet was 7.91 mm, and the length of the posterior leaflet was 5.3 mm. V-clamp type III was used to reduce regurgitation. The center of the anterior leaflet (A2) and the center of the posterior leaflet (P2) were deployed by the V-clamp device under general anesthesia. After mitral valve repair, the mean transmitral gradient was 1 mmHg. The dog was discharged 48 h after surgery. At the one-month follow-up examination, the dog exhibited remarkable clinical improvement, and the size of the LA decreased at one, two, and three months after mitral valve repair.

### 3.1. Post-Operative Clinical Data

All four patients showed clinical stability at the 150-day follow-up visit, and no patients in our study were readmitted due to decompensated heart failure. Clinical signs such as cough and dyspnea have resolved without diuretic medication. TTE revealed preserved left ventricular fractional shortening of 40.02 ± 14.24%. Echocardiography parameters such as the left atrium (LA) dimension, LA/AO ratio, left ventricular internal diameter during systole (LVIDs), mitral valve E/A ratio, mitral regurgitation volume, and cardiac output rapidly decreased within the first three days following mitral valve repair. The LA size, LA/AO ratio, LVIDs, mitral valve E/A ratio, and cardiac output subsequently gradually increased over the next five months because of residual mitral regurgitation until it stabilized at a constant value after mitral valve repair. However, the mitral regurgitation volume and cardiac output rapidly decreased within the initial three days, and then the mitral regurgitation volume gradually increased over the next five months due to residual mitral regurgitation, as shown in Figure 4. The V-clamp device remained intact and properly positioned during this period.

### 3.2. Transthoracic Echocardiography Findings After Mitral Valve Repair

The transthoracic echocardiography (TTE) findings at five and six months after the V-clamp procedure are shown in Table 3 and Figure 4, respectively. TTE at the five- and six-month follow-up visits revealed a reduced LA/AO ratio. The changes in the LA and LV internal diameters at systole were greater in patients with stage B2 disease than in those with stage C disease. The mitral valve deceleration time was shortened, and the ratio of the mitral E wave and isovolumetric relaxation time (E/IVRT) values and the pulmonary vein (PV)-to-pulmonary artery (PA) (PV/PA) ratio were reduced in most patients after surgery.

## 4. Discussion

This case series demonstrated the safety and feasibility of the V-clamp procedure for patients with stage B2 and C myxomatous mitral valve degeneration, which is a chronic condition that significantly impacts the quality of life (QOL) of dogs. Minnesotans Living with Heart Failure and Validation questionnaires were investigated in this study to evaluate the overall outcome following surgical treatment. The dogs showed improvements in respiratory signs during the follow-up visit and increased happiness and overall quality of life after the surgical intervention, as shown in Table 1. However, there were no differences in hygiene scores or physical capacity in dogs with MMVD stage C after surgery. Notably, overall QOL scores differed between dogs in MMVD stages B2 and C. Despite these findings, left ventricular function was previously observed, indicating that mitral valve repair in MMVD patients at stages B2 and C does not affect cardiac capacity. A decrease in happiness and mental status scores was observed in dogs in MMVD stage C. However, hygiene and physical capacity did not significantly differ among stage B2 dogs. These factors should be considered in further surgical treatment of these patients.

The V-clamp procedure in this study was performed on two patients with stage B2 and two patients with stage C. All four patients had an appropriate mitral valve regurgitation structure. Specifically, regurgitation was limited to the leaflet without a cleft and was primarily localized to the A2P2 position. Many MMVD patients are considered high-risk surgical patients due to severe congestive heart failure with LV dysfunction and multiple concurrent diseases, such as pancreatitis and liver or kidney dysfunction, despite mitral valve repair being a treatment option. The patient’s hydration status and levels of electrolytes, particularly potassium, sodium, calcium, and magnesium, should be monitored after the operation [17].

Transcatheter edge-to-edge repair using a V-clamp device, a procedure similar to the human MitraClip technique, is a current alternative treatment for mitral valve regurgitation in dogs and is considered an appropriate treatment for high-risk cardiac surgery patients [18,19,20]. While V-clamp procedures can improve quality of life, some potential complications include post-operative arrhythmias. The technique involves a transapical puncture with an introducer sheath to enter the left ventricle and subsequently the mitral valve. Techniques for using the V-clamp device in this study may result in acute arrhythmias during post-operative recovery. However, arrhythmias caused by cardiac muscle injury usually resolve by the healing process within a few days. Monitoring parameters such as NT-proBNP or cardiac troponin levels pre-operatively and post-operatively may help to assist in evaluating cardiac status, including cardiac stretch, dilation, and hypertrophy [21].

In our study, nonsustained ventricular tachycardia occurred in one patient following the surgical procedures. An intravenous bolus of 2 mg/kg of lidocaine (20 mg/mL) was administered and repeated every 5–10 min up to three additional times to normalize the ventricular rhythm.

Thoracic and heart surgery has been reported to be associated with significantly more pain than other surgical methods [22,23]. Pain after heart surgery is usually severe on post-operative days one and two. Despite gradual reductions in pain throughout the recovery period, most patients continue to experience severe pain. Severe pain is associated primarily with the main surgical site on postoperative day one, but from postoperative day two to postoperative day four, pain is generally caused by coughing, movement, turning, and deep breathing [24]. Local anesthesia, such as bupivacaine, has been reported to provide post-operative pain relief for up to 72 h in dogs and cats [25]. Therefore, local anesthesia is recommended for use in surgery, especially thoracic and heart surgery. Lidocaine is widely used to manage pain during and after cardiac surgery. Lidocaine might be safely administered intravenously, and intravenous infusion of lidocaine is significantly associated with reduced pain scores and decreased usage of fentanyl. However, no significant relationships were found between lidocaine administration and other parameters, such as post-operative death and length of hospital stay [26]. No data were reported for other secondary outcomes after surgery, including nausea, vomiting, or arrhythmias. Research has indicated that external sympathetic pathways play a dominant role in the nervous system of the heart. The pre-operative and post-operative administration of nerve blockers containing lidocaine has been shown to be effective at decreasing recurrent arrhythmias and alleviating the burden of ventricular arrhythmias in patients [22]. The findings of this study suggest that the effective management of acute post-operative pain, as well as electrocardiography and echocardiography assessments, are important for monitoring and stabilizing electrophysiological and heart function.

Post-operative anticoagulation management is required to prevent the formation of blood clots on catheters and devices [27,28]. In our study, antithrombotic drugs such as clopidogrel (total dose of 18.75 mg) or rivaroxaban (1 mg/kg/day) were administered for at least three months to prevent post-operative thromboembolism of the V-clamp device.

TEE evaluation is critical for accurate V-clamp device positioning, which allows for less residual mitral regurgitation after surgery. However, clamping the mitral valve might cause mitral valve stenosis. Echocardiographic parameter measurements are more challenging after the V-clamp device is put in place because clamping may cause several unequal orifices and jets, eccentric regurgitation, and acute changes in the size of the heart. Additionally, performing an electrocardiogram (ECG) record before and after surgery might help to identify any abnormalities in heart rate and rhythm, including tachycardia, bradycardia, atrial fibrillation, premature ventricular contractions, and ventricular escape beats. Continuous monitoring via ECG or Holter monitoring during the perioperative period is vital for assessing hemodynamic impact and determining the necessity for therapeutic intervention. Moreover, obtaining an accurate baseline of the patient’s heart rate and blood pressure before surgery is essential for developing an appropriate anesthetic plan and documenting cardiovascular changes throughout the surgical procedure. Direct arterial blood pressure monitoring is the gold standard for blood pressure measurement in heart surgery. Continuous monitoring of systolic arterial pressure (SAP), diastolic arterial pressure (DAP), and mean arterial pressure (MAP) via dorsal pedal artery catheterization is recommended during surgery.

The pulmonary vein (PV)-to-pulmonary artery (PA) ratio (PV/PA) has been proposed as an index that may help distinguish dogs with congestive heart failure. The normal value for this ratio is 1.0. In this study, after V-clamp device implantation, the pulmonary-vein-to-pulmonary-artery ratio (PV/PA) was closer to the normal value of 1.0, suggesting that V-clamp device implantation is another treatment option for mitral valve disease that might reduce clinical symptoms of heart failure in dogs.

In addition, in this study, after V-clamp device implantation surgery, the MV DT value decreased, indicating an increase in left atrial pressure (LAP). Additionally, an E/IVRT value greater than 2.2 may predict a mean LAP of ≥15 mmHg. Monitoring the LAP and transmitral pressure gradient during mitral valve repair provides valuable insights into immediate hemodynamic effects and post-operative clinical outcomes. The LAP is typically elevated, and reductions are typically observed within 30 days after the procedures [29,30]. Conversely, an increased LAP following mitral valve repair indicates a high risk of both post-operative heart failure and all-cause mortality. Therefore, it is advisable to carefully monitor and adjust the device position if the pressure above the mitral valve exceeds 5 mmHg [31]. However, there are limitations to our study. This single center involved a small number of patients, and the initial learning curve may have impacted the results. Additionally, the LAP measurements before and after repair were not included in the analysis. Consequently, the data for all patients were unavailable.

## 5. Conclusions

In this study, we share our perioperative experience in managing patients before and after surgery, as well as managing complications such as pain and arrhythmias. Post-operative pain management, including thrombus prevention, is important for improving patient survival after surgery.

## Figures and Tables

**Figure 1 animals-14-03068-f001:**
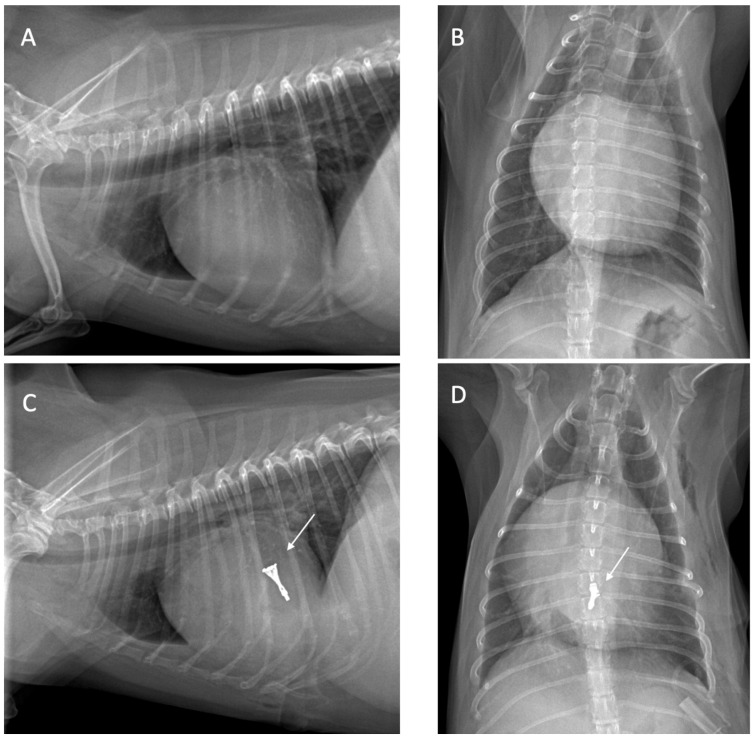
The thoracic radiograph on lateral and ventrodorsal view before (**A**,**B**) and after surgery (**C**,**D**), and the v-clamp (white arrow) was visualized on the area between the left atrium and left ventricle. The heart enlargement was visualized on the thoracic X-ray with a vertebral heart scale (VHS) of 11.5, and the vertebral left atrium size (VLAS) of 2.9.

**Figure 2 animals-14-03068-f002:**
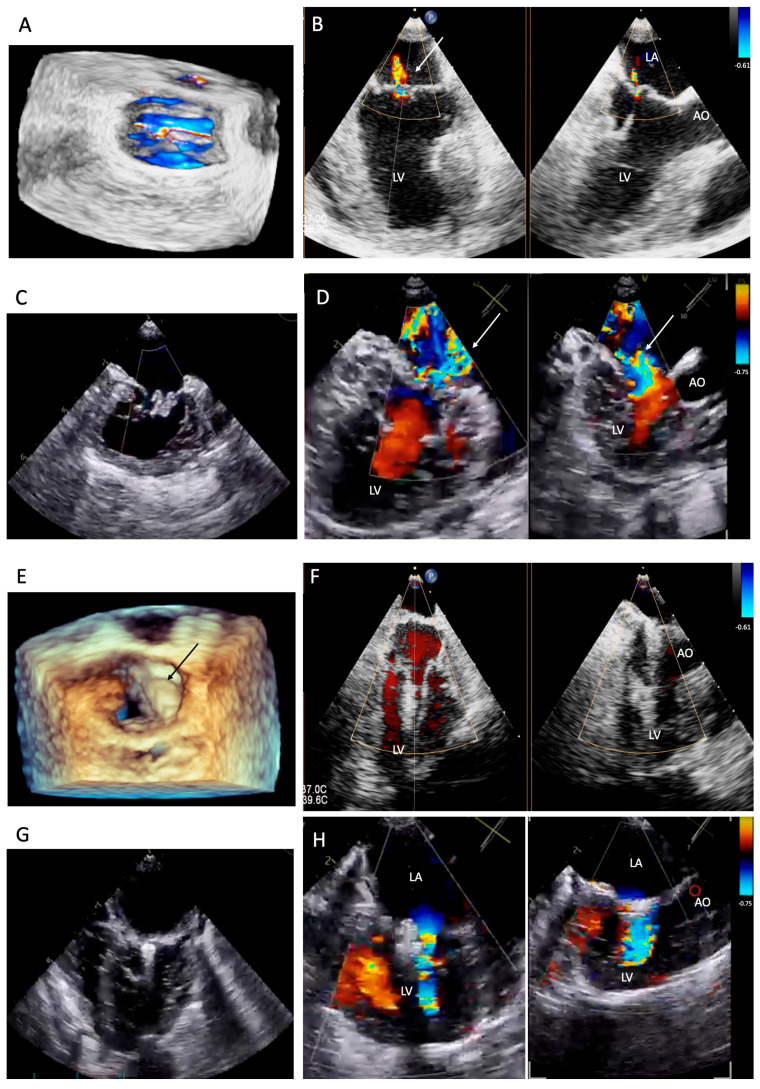
Three-dimensional transesophageal echocardiographic images from two patients before the surgical intervention (**A**–**D**); jet flow (white arrow) is visible in the left atrium (LA) (**A**,**B**,**D**). (**E**–**H**) a V-clamp device (black arrow on (**E**)) on the mitral valve of the dog; no jet flow is visible after the procedure (**F**), and slight jet flow is visible after the procedure (**H**). (LV = left ventricle; AO = aorta).

**Figure 3 animals-14-03068-f003:**
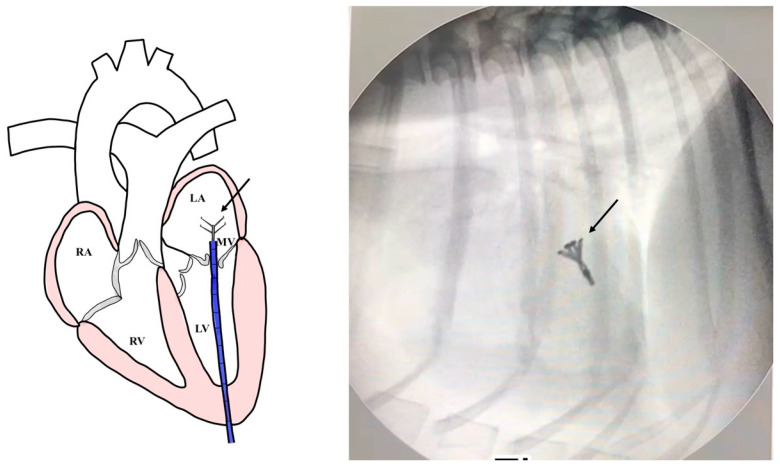
A V-clamp device placement (black arrow) with an introducer sheath of 14 mm was inserted into the left atrium (LA). A satisfactory clamp position was confirmed using fluoroscopy guidance. (RA = right atrium; RV = right ventricle; LV = left ventricle; MV = Mitral valve).

**Figure 4 animals-14-03068-f004:**
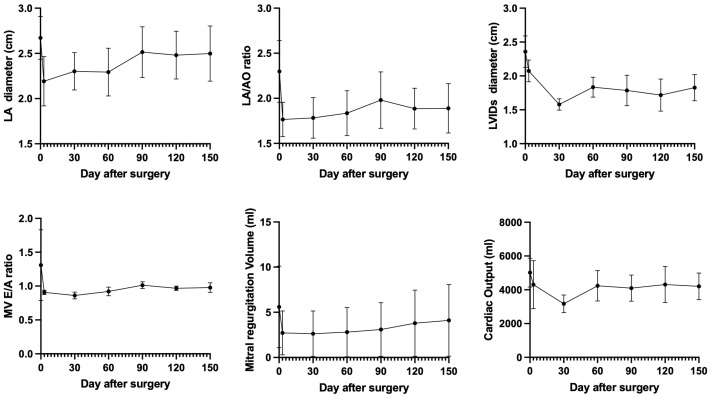
Echocardiography parameters before and after 3, 30, 60, 90, 120, and 150 days after mitral valve repair using a V-clamp device. Data were represented as mean ± standard deviation (SD).

**Table 1 animals-14-03068-t001:** Quality of life score before and three months after mitral valve repair. Each domain contains related items that were scored on a scale of 0 to 5, where 0 is the minimum level and 5 is the maximum level.

Parameter	Patient 1		Patient 2		Patient 3		Patient 4	
	Before	After	Before	After	Before	After	Before	After
Happiness								
(1) My dog wants to play.	3	4	2	3	2	2	2	4
(2) My dog reacts to my presence.	5	5	3	3	3	3	3	5
(3) My dog enjoys life.	3	4	3	3	3	3	3	5
Happiness score	11	13	8	9	8	8	8	14
Mental status								
(1) My dog shakes or trembles.	4	1	1	0	1	0	1	0
(2) My dog seems depressed and not alert.	2	2	1	1	2	1	3	0
Mental status score	6	3	2	1	3	1	4	0
Hygiene								
(1) My dog keeps him/herself clean.	4	4	3	4	2	3	4	5
(2) My dog smells like urine or has skin irritation.	0	0	0	0	1	0	0	0
(3) My dog’s hair is greasy and rough.	1	1	5	2	0	0	4	2
Hygiene score	5	5	8	6	3	3	8	7
Physical function								
(1) My dog rests and sleeps all day long.	4	4	2	2	3	2	4	1
(2) My dog lays in one place all day long.	3	3	2	1	3	2	0	0
(3) My dog moves normally.	3	4	2	3	2	2	4	5
(4) My dog is in pain.	0	0	1	1	1	1	0	0
(5) My dog coughs and pants frequently even at rest.	2	1	1	1	5	5	2	1
(6) My dog has swelling on the legs and ascites.	0	0	0	0	2	4	0	0

**Table 2 animals-14-03068-t002:** Signalment, hematological, and serum biochemical parameters before and one month after mitral valve repair.

Parameter	Patient 1		Patient 2		Patient 3		Patient 4		Reference Value
	Before	After	Before	After	Before	After	Before	After	
Age	9 years		14 years		9 years		11 years		-
Gender	Female		Male		Female		Male		-
Breed	Beagle		Pomeranian		Chihuahua		Poodle		-
MMVD stage	B2		C		C		B2		-
Vertebral heart size (VHS) [16]	11.2		11.5		12		10.5		11.1
Vertebral left atrial size (VLAS) [16]	2.5		2.9		3.1		3		2.1
AP diameter (mm)	16.34		22.8		21.7		12.7		12–24
Type of V-clamp device	IV		IV		IV		III		II–V
WBC (×10^3^/uL)	5.51	6.67	6.85	10.18	7.35	7.6	15.0	11.2	6–17
RBC (×10^6^/uL)	8.85	8.26	5.77	5.87	7.71	5.9	6.13	8.15	5–9
HGB (gm%)	16.8	18.8	13.6	13.6	17.6	14.4	12.9	14.2	12–18
PCV (%)	57.36	53.8	38.3	37.9	50.0	40	40.9	50.7	30–55
Platelets (×10^3^/uL)	232	304	261	280	416	401	215	324	200–900
Segmented neutrophil (×10^3^/uL)	3.66	6.9	5.71	8.08	8.1	8.6	8.5	8.6	3–11.5
Lymphocyte (×10^3^/uL)	1.39	2.0	7.2	6.9	1.2	1.3	0.4	0.8	1–4.8
Monocyte (×10^3^/uL)	0.38	0.9	0.6	0.6	0.2	0	0.9	0.5	0.15–1.35
Eosinophil (×10^3^/uL)	0.06	0.2	0.25	0.75	0.5	0	0.3	0.2	0.1–1.25
BUN (mg%)	13	15.7	94	44	22.2	20	16.0	37	<34.6
Creatinine (mg%)	0.9	0.88	1.7	1.1	0.9	0.6	0.7	0.7	<1.8
Total Protein (gm%)	6.5	7.4	7.8	6.1	6.8	7.0	9	7.6	5.3–7.8
ALT (U/L)	37	31	452	420	76	48	122	88	<89
ALP (U/L)	109	91	1911	877	94	85	48	45	<108

MMVD = myxomatous mitral valve degeneration; WBC = white blood cell; RBC = red blood cell; HGB = hemoglobin; PCV = pack cell volume; BUN = blood urea nitrogen; ALT = alanine transaminase; ALP = alkaline phosphatase; AP = anteroposterior dimension of the mitral valve.

**Table 3 animals-14-03068-t003:** Echocardiography parameters before and after five–six months of mitral valve repair.

Parameter	Patient 1		Patient 2		Patient 3		Patient 4	
	Before	After 6 Months	Before	After 5 Months	Before	After 6 Months	Before	After 5 Months
MMVD stage	B2		C		C		B2	
IVSd (cm)	0.84	0.87	0.70	0.62	0.70	0.55	0.74	0.81
LVIDd (cm)	4.26	3.18	3.80	3.46	3.69	3.33	2.86	2.12
LVPWd (cm)	0.72	0.93	0.80	0.60	0.60	0.56	0.56	0.68
IVSs (cm)	1.32	0.93	1.00	1.04	0.93	1.12	0.61	1.08
LVIDs (cm)	2.82	1.83	2.70	1.81	1.93	1.64	1.99	1.35
LVPWs (cm)	0.89	1.10	1.10	1.15	0.95	0.98	0.57	0.77
FS (%)	33.9	42.5	34.0	47.8	47.1	50.73	30.59	36.36
LA (cm)	2.55	2.25	3.29	3.20	2.7	2.7	2.15	1.68
LA/Ao ratio	1.44	1.60	2.95	2.0	2.74	2.39	2.06	1.61
MV E/A ratio	0.78	0.90	2.86	1.14	0.6	1.04	1.0	0.96
MV Deceleration time (ms)	80	70	100	135	85	21	140	70
E wave velocity (m/s)	0.62	0.59	1.61	1.71	1.79	1.17	0.93	1.22
E/IVRT ratio	2.07	2.91	4.02	1.98	8.95	4.68	3.64	2.4
PV/PA ratio	1.02	1.16	3.35	2.01	2.93	2.65	1.02	0.96

MV= mitral valve; MMVD = myxomatous mitral valve degeneration; IVSd = interventricular septum diastolic diameter; LVIDd = left ventricular end-diastolic diameter; LVPWd = left ventricular free wall diastolic diameter; IVSs = interventricular septum end-systolic diameter; LVIDs = left ventricular end-systolic diameter; LVPWs = left ventricular free wall end-systolic diameter; FS = left ventricular fractional shortening; LA = left atrium; LA/Ao ratio = left atrium/Aorta diameter ratio; MV E/A ratio = mitral valve E Wave/A wave velocity ratio; IVRT = isovolumetric relaxation time; PV = pulmonary vein; PA = pulmonary artery; E:IVRT = ratio of E wave velocity to isovolumetric relaxation time; PV/PA ratio = ratio of pulmonary vein to pulmonary artery.

## Data Availability

The datasets used and/or analyzed during the current study are available from the corresponding author upon reasonable request.

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
