# Peer review of "Transcatheter Edge-to-Edge Repair of the Mitral Valve in Four Dogs: Preliminary Results Regarding Efficacy and Safety"

_animals, 2024, doi:10.3390/ani14213068_

Round 1

Reviewer 1 Report

Comments and Suggestions for Authors

This paper is a case series and a valuable report on a new device, the V-clamp. However, the details of the case course are significantly insufficient, and the discussion part is not on point. If major revisions are not made appropriately, it will be difficult to accept.

Major comments

1. Add data by focusing on either perioperative management or postoperative clinical course in 4 patients. If it's the former, you should consider perioperative monitoring and anesthesia management. If it is the latter, you should consider the transition of postoperative clinical data.

2. The reason for not performing open-heart surgery should be clearly stated, along with the selection criteria for the V-clamp.

3. To clarify the therapeutic effect, please add a description or numerical value for mitral regurgitation volume.

4. Are postoperative medications discontinued from the time of discharge in all cases? Did you prescribe only clopidogrel when the patient was discharged from the hospital?

Specific comments

Line 23:Isn't there only a follow-up up of up to 4 months?

Line 120:Was the 4Fr sheath inserted directly? Have you performed a drawstring suture?

Line 121:Did you also use a 4Fr sheath to insert the V-clamp body?

Line 122:Is it the same echo device as the TEE?

Line 130 (Figure 2):4 millmeters sheath? 4Fr?

Line 131 (Figure 2):Did you use fluoroscopic guidance?

Line 133 (Figure 3):The amount of mitral regurgitation seems to be quite low, but what number is the patient?

Line 143:Was it anesthetized after surgery and evaluated by transesophageal 3D echo?

Line 151:Does it mean that the liver level rises due to congestive liver? How many cases of right heart failure did you have?

Table 3:The % notation is difficult to understand. Please fill in the actual measured value. For E/A, add the respective numbers for Ew and Aw.

Line 183:Where can I find the results of the initial 3 days?

Line 184:Was the increase over the 3 months due to residual or relapsed mitral regurgitation?

Line 192:What is cardiac support?

Line 200:ACVIM classes are basically irreversible. Specifically, describe which clinical values have improved and to what extent. Please correct it for other cases.

Line 204:Does it mean that a patient with stage C has had pulmonary edema in the past? Or there was pulmonary edema on the day of surgery?

Line 207:What device size did you use? Why doesn't Patient 1 have a numerical value such as leaflet length?

Line 232:Does it mean that there was a residual mitral regurgitation of moderate or higher? Add indicators on the regurgitation rate before and after surgery in all cases.

Line 235:B2?

Line255:Which parts of the four patients do you judge to be high-risk?

Line 259:3?

Line 270:I don't see any results about post-operative care.

Line 272:In these four patients, what arrhythmias were recognized, to what extent, and what kind of response was taken?

Line 273:I can't find any biomarker values. Consideration should be mainly based on your own results.

Line 282:Was local anesthesia administered even after surgery, for example, using a wound catheter?

Line 299:Did you do continuous monitoring using ECG or Holter monitoring?

Figure 5:Did you do direct arterial blood pressure monitoring? If so, add the data.

Comments on the Quality of English Language

It is not so much the quality of the English as the logical structure of the sentences that is inadequate and asks for correction.

Author Response

Dear Editor,

We have provided a point-by-point reply to each point raised by the reviewers as a separate file and indicated any changes or corrections within the text, highlighting the resubmitted manuscript in yellow. Please see below for the list of changes.

Answer to the reviewer:

Reviewer 1

Major comments

  1. Add data by focusing on either perioperative management or postoperative clinical course in 4 patients. If it's the former, you should consider perioperative monitoring and anesthesia management. If it is the latter, you should consider the transition of postoperative clinical data.

Answer to the reviewer:

We appreciate the helpful comment and apologize for the unclear on this point. We added the clinical results of 150 days post-operative such as LA diameter, LA/AO ratio, LVIDs, MV E/A ratio, mitral regurgitation volume, and cardiac output as shown in Figure 4.

  1. The reason for not performing open-heart surgery should be clearly stated, along with the selection criteria for the V-clamp.

Answer to the reviewer:

We apologize for the unclear criteria for V-clamp surgery. We have revised and edited the Introduction part with the reviewer’s comments. We appreciate the reviewer’s understanding.

The V-clamp is designed for canine mitral valve regurgitation and is most appropriate when regurgitation is limited to a specific leaflet, particularly when there is no leaflet cleft. The V-clamp procedure is considered lower risk than open heart surgery because it avoids the use of cardio-pulmonary bypass to stop the heartbeat and less surgical time.

The selection criteria for the V-clamp procedure that have the highest likelihood of a successful outcome include small breed dogs with ACVIM stage B2 or stage C weighing over 3 kg or the diameter of the anterior-posterior (AP) mitral valve leaflet measured on transthoracic echocardiography (TTE) over 10 mm.

  1. To clarify the therapeutic effect, please add a description or numerical value for mitral regurgitation volume.

Answer to the reviewer:

We apologize for the lack of clarity. We edited the result section (Figure 4) in the revised manuscript as reviewer’s comments. We appreciate the reviewer’s understanding.

  1. Are postoperative medications discontinued from the time of discharge in all cases? Did you prescribe only clopidogrel when the patient was discharged from the hospital?

Answer to the reviewer:

We apologize for the lack of clarity. We added this point in the revised manuscript as reviewer’s comments. We appreciate the reviewer’s understanding.

The medications, including diuretics and Pimobendan, were discontinued, and only clopidogrel was prescribed for all patients at the time of discharge. However, one dog developed right-side congestive heart failure, necessitating the re-prescription of diuretics and Pimobendan for that patient.

Specific comments

Line 23: Isn't there only a follow-up of up to 4 months?

Answer to the reviewer:

We apologize for the lack of clarity on this point. We edited the follow-up duration to 5-6 months (Table 3) in the revised manuscript as reviewer’s comments. We appreciate the reviewer’s understanding.

Line 120:Was the 4Fr sheath inserted directly? Have you performed a drawstring suture?

Answer to the reviewer:

We apologize for the mistake. We edited the method section in the revised manuscript as

reviewer’s comments. We appreciate the reviewer’s understanding.

An incision was made at the left 5th intercostal space. A 2-0 polypropylene monofilament purse-string suture was placed at the puncture area and then a 14 French percutaneous sheath introducer (HongYu Medical Technology, Shanghai, China.) was guided into the left ventricle, and the v-clamp device was inserted through the sheath and positioned in the left ventricle under transesophageal echocardiography and 3D imaging

Line 121:Did you also use a 4Fr sheath to insert the V-clamp body?

Answer to the reviewer:

We apologize for the mistake. We edited the method section in the revised manuscript as

reviewer’s comments. We appreciate the reviewer’s understanding.

a 14 French percutaneous sheath introducer (HongYu Medical Technology, Shanghai, China.) was guided into the left ventricle, and the v-clamp device was inserted through the sheath and positioned in the left ventricle under transesophageal echocardiography and 3D imaging

Line 122:Is it the same echo device as the TEE?

Answer to the reviewer:

We apologize for the lack of clarity on this point. We edited the method section in the revised manuscript as reviewer’s comments. We appreciate the reviewer’s understanding.

Pre- and postoperative TTE was performed using a General Electric Vivid 5s ultrasound system.

Perioperative 3D-TEE was performed using a Philips Epiq 7C ultrasound machine with a transducer from 1 to 7 MHz (Philips Medical Systems, Andover, MA, USA).

Line 130 (Figure 2): 4 millimeters sheath? 4Fr?

Answer to the reviewer:

We apologize for the mistake. We edited the method section (Echocardiography) in the revised manuscript as reviewer’s comments. We appreciate the reviewer’s understanding.

a 14 French percutaneous sheath introducer (HongYu Medical Technology, Shanghai, China.) was guided into the left ventricle, and the v-clamp device was inserted through the sheath.

Line 131 (Figure 2): Did you use fluoroscopic guidance?

Answer to the reviewer:

We apologize for the lack of clarity on this point. We edited the method section (Figure 2) in the revised manuscript as reviewer’s comments. We appreciate the reviewer’s understanding.

A satisfactory clamp position was confirmed using fluoroscopy and 3D TEE as shown in Figure 2.

Line 133 (Figure 3): The amount of mitral regurgitation seems to be quite low, but what number is the patient?

Answer to the reviewer:

We apologize for the lack of clarity on this point. We edited the method section (Figure 3) in the revised manuscript as reviewer’s comments. We appreciate the reviewer’s understanding.

Patient 1 has a mild mitral valve regurgitation with LA slightly dilation.

Line 143:Was it anesthetized after surgery and evaluated by transesophageal 3D echo?

Answer to the reviewer:

We apologize for the lack of clarity on this point. We edited the method section in the revised manuscript as reviewer’s comments. We appreciate the reviewer’s understanding.

 Postoperative evaluation was performed using transthoracic echocardiography.

Line 151:Does it mean that the liver level rises due to congestive liver? How many cases of right heart failure did you have?

Answer to the reviewer:

We apologize for the lack of clarity on this point. We edited the method section in the revised manuscript as reviewer’s comments. We appreciate the reviewer’s understanding.

One patient had moderate tricuspid valve regurgitation with raised pulmonary artery pressures, after surgery, the patient developed right-side congestive heart failure, necessitating the re-prescription of diuretics and Pimobendan for the patient.

Table 3: The % notation is difficult to understand. Please fill in the actual measured value. For E/A, add the respective numbers for Ew and Aw.

Answer to the reviewer:

We apologize for the lack of clarity on this point. We edited Table 3 in the revised manuscript as reviewer’s comments. We appreciate the reviewer’s understanding.

Line 183:Where can I find the results of the initial 3 days?

Answer to the reviewer:

We apologize for the lack of clarity on this point. The results of the initial 3 days are represented in Figure 4. We appreciate the reviewer’s understanding. 

Line 184:Was the increase over the 3 months due to residual or relapsed mitral regurgitation?

Answer to the reviewer:

We apologize for the lack of clarity on this point. We edited the discussion section to clarify this point in the revised manuscript as reviewer’s comments. We appreciate the reviewer’s understanding.

Echocardiography parameters such as the left atrium (LA) dimension, LA/AO ratio, left ventricular internal diameter during systole (LVIDs), Mitral valve E/A ratio, Mitral regurgitation volume, and cardiac output showed a rapid decrease within the initial 3 days following mitral valve repair. Subsequently, the LA size, LA/AO ratio, LVIDs, Mitral valve E/A ratio, and cardiac output exhibited a gradual increase over the next 4-5 months due to residual mitral regurgitation until it stabilized at a constant value after mitral valve repair. However, Mitral Regurgitation volume and cardiac output showed a rapid decrease within the initial 3 days, and then the Mitral Regurgitation volume gradually increased over the next 4-5 months due to residual mitral regurgitation as shown in Figure 4.

Line 192:What is cardiac support?

Answer to the reviewer:

We apologize for the lack of clarity on this point. We edited the revised manuscript as reviewer’s comments. We appreciate the reviewer’s understanding.

The cardiac support includes co-enzyme Q10 and L-carnitine.

Line 200:ACVIM classes are basically irreversible. Specifically, describe which clinical values have improved and to what extent. Please correct it for other cases.

Answer to the reviewer:

We apologize for the mistake. We edited the results section in the revised manuscript as reviewer’s comments. We appreciate the reviewer’s understanding.

Line 204:Does it mean that a patient with stage C has had pulmonary edema in the past? Or there was pulmonary edema on the day of surgery?

Answer to the reviewer:

We apologize for the lack of clarity on this point. We edited the revised manuscript as reviewer’s comments. We appreciate the reviewer’s understanding.

One patient with stage C has had pulmonary edema in the past.

Line 207:What device size did you use? Why doesn't Patient 1 have a numerical value such as leaflet length?

Answer to the reviewer:

We apologize for the lack of clarity on this point. We edited the evaluation part and Table 2 in the revised manuscript as reviewers’ comments to clarify this point. We appreciate the reviewer’s understanding.

Line 232:Does it mean that there was a residual mitral regurgitation of moderate or higher? Add indicators on the regurgitation rate before and after surgery in all cases.

Answer to the reviewer:

We would like to thank the reviewer for this helpful advice to strengthen the manuscript. In the revised manuscript, we've added the indicators on the regurgitation rate, including MR Vmax, MR VTI, MR PISA, MR CSA, and MR stroke volume before and after surgery in all cases, as suggested by the reviewer. 

Line 235:B2?

Answer to the reviewer:

We apologize for the mistake. We edited 2 to B2 in the revised manuscript. We appreciate the reviewer’s understanding.

Line255:Which parts of the four patients do you judge to be high-risk?

Answer to the reviewer:

We apologize for the lack of clarity on this point. We edited the discussion part and Table 2 in the revised manuscript as reviewers’ comments to clarify this point. We appreciate the reviewer’s understanding.

Line 259:3?

Answer to the reviewer:

We apologize for the mistake. We edited 3 to [3] in the revised manuscript. We appreciate the reviewer’s understanding.

Line 270:I don't see any results about post-operative care.

Answer to the reviewer:

We would like to thank the reviewer for this helpful advice to strengthen the manuscript. We closely monitored all dogs in this study for potential adverse effects associated with the operation. This monitoring was conducted at regular intervals, such as after surgery and monthly, and included pain management, physical examination, chest X-ray, blood profile analysis, electrocardiography, and echocardiographic study. In the revised manuscript, we have clarified this point and included a paragraph about post-operative care in the material and methods, as suggested by the reviewer.

Line 272:In these four patients, what arrhythmias were recognized, to what extent, and what kind of response was taken?

Answer to the reviewer:

We apologize for the lack of clarity on this point. We edited the discussion part in the revised manuscript as reviewers’ comments to clarify this point. We appreciate the reviewer’s understanding.

In this study, nonsustained ventricular tachycardia occurred following the surgical procedures. However, the arrhythmias resolved within a few days as part of the healing process.

Line 273:I can't find any biomarker values. Consideration should be mainly based on your own results.

Answer to the reviewer:

We apologize for the lack of clarity on this point.

Because the four patients in this study did not measure the biomarker, and one patient had arrhythmias after surgery. So, biomarker measurements such as NT-proBNP or cardiac troponin levels were suggested in the discussion section.

We edited the discussion part in the revised manuscript as reviewers’ comments to clarify this point. We appreciate the reviewer’s understanding.

The most common cause of ventricular tachycardia is coronary artery obstruction, which reduces blood flow to the heart due to cardiac muscle damage. In our study, nonsustained ventricular tachycardia occurred in one patient following the surgical procedures. However, the arrhythmias resolved within a few days as part of the healing process. If ventricular tachycardia lasts for more than 30 seconds or becomes sustained, it can lead to ventricular fibrillation, cardiac arrest, or sudden cardiac death. Therefore, monitoring parameters such as NT-proBNP or cardiac troponin levels preoperatively and postoperatively may help to assist in evaluating the cardiac status, including cardiac stretch, dilation, and hypertrophy.

Line 282:Was local anesthesia administered even after surgery, for example, using a wound catheter?

Answer to the reviewer:

We apologize for the lack of clarity on this point.

All 4 dogs in this study, had an intercostal block with bupivacaine (at ribs 5, 6, 7, 8, and 9) for pre-operation. However, due to the bupivacaine duration of action, we did not perform a local block after completion of surgery. In addition, a chest drain was placed in all 4 patients until the day they were discharged from the hospital.

Line 299:Did you do continuous monitoring using ECG or Holter monitoring?

Answer to the reviewer:

We would like to thank the reviewer for this helpful advice to strengthen the manuscript. All dogs in this study underwent continuous ECG monitoring at regular monthly intervals following an operation.

Figure 5: Did you do direct arterial blood pressure monitoring? If so, add the data.

Answer to the reviewer:

We would like to thank the reviewer for this helpful advice to strengthen the manuscript. In the revised manuscript, we've added the direct arterial blood pressure data as suggested by the reviewer. 

Comments on the Quality of English Language

It is not so much the quality of the English as the logical structure of the sentences that is inadequate and asks for correction.

Answer to the reviewer:

We would like to thank the reviewer for this helpful advice to strengthen the manuscript. We’ve checked the English as the logical structure of the sentences thoroughly in the revised manuscript as suggested by the reviewer.  However, we have asked an English native speaker who has experience in writing manuscripts in biomedical science to extensively proofread the revised manuscript as suggested by the reviewer.  We believe that the revised manuscript has been much improved regarding the English writing.

Best Regards

Soontaree Petchdee

Reviewer 2 Report

Comments and Suggestions for Authors

I would like to thank the authors for the time and effort dedicated to write this manuscript. And also, for their work on this disease that affects so many dogs worldwide and that we lack a cure for. Any new suitable options for its treatment like the v-clamp are good news. 

Having said that I would like to make some comments, suggestions, recommendations about this manuscript. 

The information given is not well presented, most of the information given on the discussion section should go on the introduction. The introduction lacks information and the objectives of this work are not mentioned. The information about the patients presented on the results section should go on the materials and methods. Materials and methods sections lacks information, what veterinary center did the dogs attended?, you mention that the study was conducted at the faculty of Veterinary Medicine but under the laboratory investigations epigraph, what was their clinical condition, why were those animals selected for the procedure?; more information on the surgical procedure which is what this paper is about would add valuable information to the paper. The results should relate to the objectives of the study, and so should the conclusion. 

Line 12 This technique... What technique?, the reader does not need to know the name of the technique. Simple summary does not mean lame, vague.  Authors should refer to the author guidelines on how a simple summary needs to be. 

Line 18-20 dogs that underwent mitral valve repair between December 2023 and June 2024.

Later authors mention that dogs were re-examined 3 months after surgery (Line 81), and after 4 months (Line 177). It is July 2024, when was the last surgery performed? Long-term follow-up (line 18) after 4 months, is it actually long-term?

Line 23 Echocardiography was conducted at baseline and the 6-month follow-up. 

The information about periods of time is misleading, needs clarifying throughout the whole manuscript. 

Line 33-34 The authors should clarify that thoes mitral valve leaflet characteristics refer to a diseased mitral valve. 

Line 36 medication slows the heart remodeling process. It does not slows the progression of heart failure, but delays it presentation. 

Line 39-40 Some linking words are needed to give sense to the sentence. 

Lack of information on the introduction. 

Line 54. Lack of information. A first paragrap should address patient information, as mentioned before. 

Line 66. Were the dogs sedated for the x-rays. What x-ray machine was used, was analog, digital, direct, indirect? What information other that device placement did you obtain? Authors mention later on the study, Table 2, VHS and VLAS measurement, this should be mentioned on materials and methods. Information regarding signs of congestive heart failure after the procedure, pulmonary edema, pleural effusion, or other like device migration, pericardial effusion would be part of the radiographic examination pre and post-op, this would add valuable information for the reader. 

The echocardiography epigraph needs re-writting. Authors should mention the probe/probes used and the manufacturer of the ultrasound machine. The information should be organized in and orderly manner, patient positioning, windows used, information look for on the different views. Acronyms in brackets after the full name. 

Line 89 Ventricular wall thickness and dimensions... I think it should say ventricular wall thickness and chamber dimension. 

Pulse wave Doppler is an ultrasound technique, is not an echocardiography. 

Line 101 The diastolic function indicated by the E-wave per A-wave ratio was performed in the left apical... This is not well expressed, the diastolic function is assessed would be more appropiate. 

In general, it is recommended to mention the name of the dug used and the manufacturing laboratory, and concentration. 

Line 118 Surgical intervention. This needs further explanation as mentioned previously. For instance, authors say that 'the chest drainage and the urinary catheter were removed', but you forget to say when and why were they set in the first place. How many days in hospital after the procedure. Explain the goal of daily radiographic examinations, as recommended before. 

On Table 2, patients 1 and 4 are classified as Stage B2 but later on the text, line 191 and 253 they were on diuretics. According to ACVIM guidelines, that you mention, Stage B2 dogs are not treated with diuretics, neither furosemide nor spironolactone.

Line 137 Evaluation. This paragraph is confusing. It the reason of it is to present the patients it should go at the beginning of Materials and Methods (M&M). Why the patients were selected, clinical signs, and if on medication. Medication is mentioned later on results, it should go on M&M. You say that thoracic radiography revealed left heart enlargement, this is part of the clinical status and should go with the patient information. This assumption of left heart enlargement on radiographs was based on what signs. Authors should refer here to VHS and VLAS not only on Table 2.

Line 142 Figure 2 is the representation of the V-clamp device and a fluoroscopic image. Figure 1, shows a dog with signs of heart enlargement. Authors should refer to that figure and add that information for the readers based on cardiac measurements, for instance. 

The Minnesotans living with heart failure and validation questionnaire should be introduce on the introduction section, the results gathered from them should be include in results, and commented in the discussion, it authors consider it an objective of the study. I do not think it is relevant in this study the way it is presented, and if not part of the objectives it should not be inlcuded, alongside Table 1 which is not relevant. 

Line 152-153 This information should go on Results. 

Table 3 should also show the raw data for the last measurements, not just the change percentage.

Line 180 the fraction of shortening value is a media and SD of the four dogs?

Line 185 You say that chambers dimensions decrease until four months that they stabilize at a constant value. To make that affirmation I would need at least another measurement after the four, and confirme that tendency. 

All the information regarding patients before surgery should go on M&M section.

Line 204 transthoracic echocardiographyu revealed Stage C by heart remodeling. Stage C is a clinical stage with signs of congestive heart failure, at present or past. Transthoracic pulmonary echocardiography can help with the identification of B-lines compatible with pulmonary edema, or echocardiography with indirect signs of congestive heart failure, for example: PV/PA>1,7, E/A>2, E wave>1,2m/s, E/IVRT>2,5, E/E'>11,7.    

Line 207 AP, what does it means?

Line 216 LA decreased indicating improvement. Line 230 left ventricle dimension increase indicating clinical improvement. Could you explaine this more acurately?

Line 217 A canine patient in Stage C remains in Stage C although they do not have congestive heart failure signs. However, this might need a redefinition with the developing of this new mitral valve repairing techniques. 

Line 221 as line 204.

Line 224 as line 207.

In line 332 you mentioned the small number of patients, this precludes a statistical analysis of results, it should be mention. 

Discussion section needs rewritting. In this section authors must comment theirs results compared to similar results from other studies, and if there are not other studies to compare with, which is the case, address the relevance of their results according to the features assess, decrease/increase in heart chamber remodeling and why, side effects of the procedure observed, why they can happen and how to approach or avoid them... Nothing is mentioned on the use of antithrombotic therapy on this paper patients but it is part of the discussion. 

The conclusion can be just sharing one experience. According to your manuscript only one patient suffer an arrythmic episode and pain. The conclusion refers to the objectives of the study, but they are not clarify on the text. I figure they would be the safety of the procedure, outcome of the patients, reversing of cardiac remodeling and few side effects. 

One of the references used by the authors could be used as a guide as to how to write and present data on a manuscript, reference number 12. This paper is similar to the study present here but with healthy dogs.

Sasaki, K., Ma, D., Mandour, A. S., Ozai, Y., Yoshida, T., Matsuura, K., Takeuchi, A., Cheng, C. J., El-Husseiny, 396 H. M., Hendawy, H., Shimada, K., Hamabe, L., Uemura, A., & Tanaka, R. (2021). Evaluation of Changes in the 397 Cardiac Function before and after Transcatheter Edge-to-Edge Mitral Valve Repair in Healthy Dogs: Conven-398 tional and Novel Echocardiography. Animals : an open access journal from MDPI, 12(1), 56. 399 https://doi.org/10.3390/ani12010056.

Comments on the Quality of English Language

Overall, language is fine but some paragraphs needs rewriting, use of appropiate and correct positioning of words. 

Author Response

Dear Editor,
We have provided a point-by-point reply to each point raised by the reviewers as a separate file and indicated any changes or corrections within the text, highlighting the resubmitted manuscript in yellow. Please see below for the list of changes. 
Reviewer 2
Answer to the reviewer: 
I would like to thank the authors for the time and effort dedicated to write this manuscript. And also, for their work on this disease that affects so many dogs worldwide and that we lack a cure for. Any new suitable options for its treatment like the v-clamp are good news. 

Having said that I would like to make some comments, suggestions, recommendations about this manuscript. 
The information given is not well presented, most of the information given on the discussion section should go on the introduction. The introduction lacks information and the objectives of this work are not mentioned. The information about the patients presented on the results section should go on the materials and methods. Materials and methods sections lacks information, what veterinary center did the dogs attended?, you mention that the study was conducted at the faculty of Veterinary Medicine but under the laboratory investigations epigraph, what was their clinical condition, why were those animals selected for the procedure?; more information on the surgical procedure which is what this paper is about would add valuable information to the paper. The results should relate to the objectives of the study, and so should the conclusion. 
Answer to the reviewer: 
We would like to thank the reviewer for his/her positive comment on the significance of our findings and would like to thank the reviewer for this helpful advice to strengthen the manuscript.  We’ve revised and checked that all comments have been clarified in the revised manuscript as suggested by the reviewer. Below are our responses to the reviewer’s suggestions

Line 12 This technique... What technique?, the reader does not need to know the name of the technique. Simple summary does not mean lame, vague.  Authors should refer to the author guidelines on how a simple summary needs to be. 
Answer to the reviewer: 
We would like to thank the reviewer for this helpful advice to strengthen the manuscript. We apologized and agreed with the reviewer on this point. However, we have cited and added references to previous guidelines in the revised manuscript, as suggested by the reviewer.

Line 18-20 dogs that underwent mitral valve repair between December 2023 and June 2024.
Later authors mention that dogs were re-examined 3 months after surgery (Line 81), and after 4 months (Line 177). It is July 2024, when was the last surgery performed? Long-term follow-up (line 18) after 4 months, is it actually long-term?
Answer to the reviewer: 
We apologize for the lack of clarity on this point. We've added this point as suggested by the reviewer.
Four patients underwent mitral repair between December 2023 and July 2024.  The last patient had surgery in early March 2024. The dogs were followed up monthly for five- and six-months post-operation. A questionnaire was completed by the owners before the surgery and three months afterward.

Line 23 Echocardiography was conducted at baseline and the 6-month follow-up. 
The information about periods of time is misleading, needs clarifying throughout the whole manuscript. 
Answer to the reviewer: 
We would like to thank the reviewer for this helpful advice to strengthen the manuscript. We have added information about periods of time throughout the revised manuscript as suggested by the reviewer.

Line 33-34 The authors should clarify that those mitral valve leaflet characteristics refer to a diseased mitral valve. 
Answer to the reviewer: 
We apologize for the lack of clarity on this point. We've added this point as suggested by the reviewer.
The mitral valve is composed of two leaflets, the anterior and posterior leaflets. Both leaflets are divided into three zones: A1, A2, A3, and P1, P2, and P3, respectively. Abnormal characteristics of the mitral valve leaflet include elongated chordae tendinae, thickening, and shrinkage of the leaflets with bulging or prolapse toward the left atrium.

Line 36 medication slows the heart remodeling process. It does not slows the progression of heart failure, but delays it presentation. 
Answer to the reviewer: 
We would like to thank the reviewer for this helpful advice to strengthen the manuscript. We have changed this sentence in the revised manuscript as suggested by the reviewer.
Line 39-41, the statement now reads, "Medication is the standard treatment for this disease, which delays the presentation of heart failure and the heart remodeling process."

Line 39-40 Some linking words are needed to give sense to the sentence. 
Answer to the reviewer: 
We would like to thank the reviewer for this helpful advice to strengthen the manuscript. As suggested by the reviewers, we added linking words and revised the sentence in the manuscript. 
Line 41-42, the statement now reads, "Therefore, common treatment practices and ACVIM guidelines recommend individualized medication treatment.”
Lack of information on the introduction. 
Answer to the reviewer: 
We would like to thank the reviewer for this helpful advice to strengthen the manuscript. In the introduction part, we have added the information on the V-clamp procedure as suggested by the reviewers.
Line 56-63, the statement now reads, "The V-clamp is designed for canine mitral valve regurgitation and is most appropriate when regurgitation is limited to a specific leaflet, particularly when there is no leaflet cleft. The V-clamp procedure is considered lower risk than open heart surgery because it avoids the use of cardio-pulmonary bypass to stop the heartbeat and less surgical time.
The selection criteria for the V-clamp procedure that have the highest likelihood of a successful outcome include small breed dogs with ACVIM stage B2 or stage C weighing over 3 kg or the diameter of the anterior-posterior (AP) mitral valve leaflet measured on transthoracic echocardiography (TTE) over 10 mm.”

Line 54. Lack of information. The first paragraph should address patient information, as mentioned before. 
Answer to the reviewer: 
We would like to thank the reviewer for this helpful advice to strengthen the manuscript. We have added the patient information in the revised manuscript as suggested by the reviewers.

Line 197-233, the statement now reads, “Patient 1: A nine-year-old female Beagle dog with a previous history of cough and evidence of ACVIM stage B2 in May 2023. Before the mitral valve repair, her prescription included 0.25 mg/kg pimobendane q12h, 1.0 mg/kg daily spironolactone, 2.0 mg/kg daily furosemide, and cardiac support including co-enzyme Q10 and L-carnitine. Transthoracic echocardiography revealed stage B2 characterized by LA dilation and mild mitral valve regurgitation. 3D transesophageal echocardiography was used to observe the main regurgitation orifice. Under general anesthesia, the center of the anterior leaflet (A2) and the center of the posterior leaflet (P2) were approximated using the v-clamp device to reduce the regurgitation. 
Patient 2: A fourteen-year-old female Pomeranian dog with a previous history of cough. Before the mitral valve repair, his prescription included 0.25 mg/kg pimobendane q12h, 1.0 mg/kg daily spironolactone, 2.0 mg/kg furosemide q12h, cardiac support including co-enzyme Q10 and L-carnitine, and liver support. Transthoracic echocardiography revealed stage C, characterized by LA and LV dilation and severe mitral valve regurgitation. 3D transesophageal echocardiography was used to observe the main regurgitation orifice. The anterior leaflet of the mitral valve was prolapsed with a 22.8 anteroposterior (AP) diameter, The length of the anterior leaflet was 13.2 millimeters, and the length of the posterior leaflet was 9.9 millimeters. 
Patient 3: A nine-year-old female chihuahua dog with a previous history of cough and increased respiratory effort. Before mitral valve repair, her prescription included 0.25 mg/kg pimobendane q12h, 1.0 mg/kg daily spironolactone, and 2.0 mg/kg furosemide q12h and cardiac support including co-enzyme Q10 and L-carnitine. Transthoracic echocardiography revealed stage C disease, characterized by LA and LV dilation and severe mitral valve regurgitation, moderate tricuspid valve regurgitation with raised pulmonary artery pressures. 3D transesophageal echocardiography was used to observe the main regurgitation orifice. No flail of the mitral valve was observed with a 21.7 AP diameter. The length of the anterior leaflet was 7.7 millimeters, and the length of the posterior leaflet was 5.8 millimeters. Under general anesthesia, the center of the anterior leaflet (A2) and the center of the posterior leaflet (P2) were deployed by the v-clamp device to reduce regurgitation. 
Patient 4: An eleven-year-old male Poodle dog with a previous history of cough and evidence of ACVIM stage B2 in April 2022. Before mitral valve repair, his prescription included 0.25 mg/kg pimobendane q12h, 1.0 mg/kg daily spironolactone, 2.0 mg/kg daily furosemide, and cardiac support including co-enzyme Q10 and L-carnitine. Transthoracic echocardiography revealed stage B2 disease, characterized by LA dilation and moderate mitral valve regurgitation. 3D transesophageal echocardiography was used to observe the main regurgitation orifice. The anterior leaflet of the mitral valve was prolapsed with a 12.7 AP diameter. The length of the anterior leaflet was 7.91 millimeters, and the length of the posterior leaflet was 5.3 millimeters.”

Line 66. Were the dogs sedated for the x-rays. What x-ray machine was used, was analog, digital, direct, indirect? What information other that device placement did you obtain? Authors mention later on the study, Table 2, VHS and VLAS measurement, this should be mentioned on materials and methods. Information regarding signs of congestive heart failure after the procedure, pulmonary edema, pleural effusion, or other like device migration, pericardial effusion would be part of the radiographic examination pre and post-op, this would add valuable information for the reader. 
Answer to the reviewer: 
We would like to thank the reviewer for this helpful advice to strengthen the manuscript. We have revised the information on the x-ray in the Materials and Methods as suggested by the reviewers.
Line 80-86, the statement now reads, "All radiographs were obtained using a digital radiography system, with dogs positioned in right lateral recumbency, and dorsoventral views. Thoracic radiographic images were captured using a GE Revolution XR/d digital X-ray system, operated at tube potentials from 60 to 130 kVp. The lung field, dilation of the pulmonary artery and vein, elevation of the distal part of the trachea toward the spine, and the presence of cardiomegaly were assessed. The size of the heart and left atrium were determined using the vertebral heart scale (VHS) and vertebral left atrium size (VLAS).”

The echocardiography epigraph needs re-writting. Authors should mention the probe/probes used and the manufacturer of the ultrasound machine. The information should be organized in and orderly manner, patient positioning, windows used, information look for on the different views. Acronyms in brackets after the full name. 
Answer to the reviewer: 
We would like to thank the reviewer for this helpful advice to strengthen the manuscript. We have revised the information of the echocardiographic study in the Materials and Methods as suggested by the reviewers.
Line 100-136, the statement now reads, “A probe size of 6 MHz was used in this study. Echocardiography was performed at baseline, and every month until 5-6 months after the surgical intervention by two skilled sonographers. The measurement was performed in a right parasternal long axis and right parasternal short axis in a right lateral recumbent position, whereas and left apical four-chamber view was performed in a left lateral recumbent position without sedation. Echocardiographic images were captured and stored for offline analysis.
In the right parasternal long and short axis view, left ventricular wall structure and function were calculated by measuring the images from two-dimensional and M-mode planes. LV wall thickness, LV dimension, and LV function were evaluated by M-mode echocardiography in the right parasternal short axis (at the base of the heart and the level of the papillary muscles) [10].  Ventricular wall thickness and dimensions were recorded during diastole and systole to obtain the parameters such as diastolic interventricular septum thickness (IVSd), systolic interventricular septum thickness (IVSs), left ventricular end-diastolic diameter (LVIDd), left ventricular end-systolic diameter (LVIDs), left ventricular wall diastolic thickness (LVPWd) and left ventricular wall systolic thickness (LVPWs). All averaged M-mode chamber measurements were normalized by body weight using the Cornell allometric scale for dogs [11].  In addition, M-mode echocardiography parameters were used to calculate the percentage of fractional shortening (%FS) and percentage ejection fraction (%EF) by using the Teicholz formula which was accomplished automatically by the echocardiographic equipment software. The right parasternal short-axis view was used to measure the left atrial dimension (LA) and aortic dimension (AO) in early diastole. Then the left atrial to aortic root ratio (LA: AO ratio) was calculated using the Swedish method. Three consecutive beats of cardiac cycles were measured, and the average values were used for all echocardiographic parameters. Right ventricular dimensions and pulmonary artery diameters were observed. 
In the left apical four-chamber view, the diastolic function indicated by the E-wave per A-wave ratio was performed in the left apical four-chamber view using a pulse wave Doppler echocardiography. Pulmonary velocity was evaluated using a pulse wave Doppler echocardiography. Moreover, we investigated and calculated the left ventricular outflow tract (LVOT) and mitral regurgitation (MR) rates to determine the LVOT stroke volume and the MR stroke volume before and after surgery in all cases.
Perioperative 3D- transesophageal echocardiogram (TEE) was performed using a Philips Epiq 7C ultrasound machine with a transducer from 1 to 7 MHz (Philips Medical Systems, Andover, MA, USA). TEE was obtained to plan the procedure and confirm the appearance of the heart valves, including the leak location. A TEE evaluation is critical for accurate V-clamp device positioning, which allows for less residual mitral regurgitation after surgery. However, clamping the mitral valve can cause mitral valve stenosis. Echocardiographic parameters measurements are more challenging after the V-clamp device is placed because clamping may cause several unequal orifices and jets, eccentric regurgitation, and acute changes in the size of the heart.”

Line 89 Ventricular wall thickness and dimensions... I think it should say ventricular wall thickness and chamber dimension. 
Answer to the reviewer: 
We would like to thank the reviewer for this helpful advice to strengthen the manuscript. We have revised the sentence as suggested by the reviewers.
Line 110-114, the statement now reads, “Ventricular wall thickness and chambers dimensions were recorded during diastole and systole to obtain the parameters such as diastolic interventricular septum thickness (IVSd), systolic interventricular septum thickness (IVSs), left ventricular end-diastolic diameter (LVIDd), left ventricular end-systolic diameter (LVIDs), left ventricular wall diastolic thickness (LVPWd) and left ventricular wall systolic thickness (LVPWs).”

Pulse wave Doppler is an ultrasound technique, is not an echocardiography. 
Answer to the reviewer: 
We would like to thank the reviewer for this helpful advice to strengthen the manuscript. We have revised the sentence in the revised manuscript as suggested by the reviewers.
Lines 123-125, the statement now reads, “In the left apical four-chamber view, the diastolic function indicated by the E-wave per A-wave ratio was performed in the left apical four-chamber view using a pulse wave (PW) Doppler technique. Pulmonary velocity was evaluated using a PW Doppler technique.”

Line 101 The diastolic function indicated by the E-wave per A-wave ratio was performed in the left apical... This is not well expressed, the diastolic function is assessed would be more appropiate. 
Answer to the reviewer: 
We would like to express our gratitude to the reviewer for providing valuable advice that has strengthened the manuscript. We apologized to the reviewer for not investigating the diastolic function using the Tissue Doppler Imaging (TDI) technique. However, we have added more mitral inflow parameters, such as MV deceleration time (DT), and E wave velocity, in the revised manuscript (In table 3) to express more information about the diastolic function as suggested by the reviewer.

In general, it is recommended to mention the name of the drug used and the manufacturing laboratory, and concentration. 
Answer to the reviewer: 
We apologize for the lack of clarity on this point. We've added this point as suggested by the reviewer.
Line 118 Surgical intervention. This needs further explanation as mentioned previously. For instance, authors say that 'the chest drainage and the urinary catheter were removed', but you forget to say when and why were they set in the first place. How many days in hospital after the procedure. Explain the goal of daily radiographic examinations, as recommended before. 
Answer to the reviewer: 
We apologize for the lack of clarity on this point. We've added this point as suggested by the reviewer.
A chest drainage tube and a urinary catheter were placed after completing the surgical procedure. Chest drainage is used for the removal of air, fluid, or blood. Postoperative urine monitoring is also important for identifying any post-operative complications. Additionally, blood profiles and chest X-rays should be evaluated daily until the patient is discharged from the hospital.

On Table 2, patients 1 and 4 are classified as Stage B2 but later on the text, line 191 and 253 they were on diuretics. According to ACVIM guidelines, that you mention, Stage B2 dogs are not treated with diuretics, neither furosemide nor spironolactone.
Answer to the reviewer: 
We apologize for the lack of clarity on this point. We appreciate the reviewer’s understanding.
The patient was referred from many hospitals, and this was the patient's original treatment before undergoing the surgery. However, in all 4 patients, the medications, including diuretics and Pimobendan, were discontinued, and only clopidogrel was prescribed for all patients at the time of discharge. However, one dog developed right-side congestive heart failure, necessitating the re-prescription of diuretics and Pimobendan for that patient.

Line 137 Evaluation. This paragraph is confusing. It the reason of it is to present the patients it should go at the beginning of Materials and Methods (M&M). Why the patients were selected, clinical signs, and if on medication. Medication is mentioned later on results, it should go on M&M. You say that thoracic radiography revealed left heart enlargement, this is part of the clinical status and should go with the patient information. This assumption of left heart enlargement on radiographs was based on what signs. Authors should refer here to VHS and VLAS not only on Table 2.
Answer to the reviewer: 
We apologize for the lack of clarity on this point. We've edited the revised manuscript as suggested by the reviewer.
Line 142 Figure 2 is the representation of the V-clamp device and a fluoroscopic image. Figure 1, shows a dog with signs of heart enlargement. Authors should refer to that figure and add that information for the readers based on cardiac measurements, for instance. 
Answer to the reviewer: 
We apologize for the lack of clarity on this point. We've edited the revised manuscript as suggested by the reviewer.

The Minnesotans living with heart failure and validation questionnaire should be introduce on the introduction section, the results gathered from them should be include in results, and commented in the discussion, it authors consider it an objective of the study. I do not think it is relevant in this study the way it is presented, and if not part of the objectives it should not be inlcuded, alongside Table 1 which is not relevant. 
Answer to the reviewer: 
We apologize for the lack of clarity on this point. 
The Minnesotans living with heart failure and validation questionnaire has been investigated in this study to see the overall outcome after surgical treatment. However, I have added discussion about this point to the discussion part, as suggested by the reviewer. We appreciate the reviewer’s understanding.

Line 152-153 This information should go on Results. 
Answer to the reviewer: 
We apologize for the lack of clarity on this point. We've moved the sentence to the result section as suggested by the reviewer.

Table 3 should also show the raw data for the last measurements, not just the change percentage.
Answer to the reviewer: 
We apologize for the lack of clarity on this point. We edited Table 3 in the revised manuscript as reviewer’s comments. We appreciate the reviewer’s understanding.

Line 180 the fraction of shortening value is a media and SD of the four dogs?
Answer to the reviewer: 
We apologize for the lack of clarity on this point. Data were represented as mean ± standard deviation (SD). The X column represents time, and the response at each time point is entered in 4 patients for the Echocardiography parameters using GraphPad Prism 9 software.

Line 185 You say that chambers dimensions decrease until four months that they stabilize at a constant value. To make that affirmation I would need at least another measurement after the four, and confirme that tendency. 
Answer to the reviewer: 
We appreciate the thoughtful comment and apologize for the lack of clarity. We have edited the revised manuscript. (Yellow highlight) We appreciate the reviewer’s understanding.

All the information regarding patients before surgery should go on M&M section.
Answer to the reviewer: 
We apologize for the lack of clarity. We have edited the revised manuscript. (Yellow highlight) We appreciate the reviewer’s understanding.

Line 204 transthoracic echocardiographyu revealed Stage C by heart remodeling. Stage C is a clinical stage with signs of congestive heart failure, at present or past. Transthoracic pulmonary echocardiography can help with the identification of B-lines compatible with pulmonary edema, or echocardiography with indirect signs of congestive heart failure, for example: PV/PA>1.7, E/A>2, E wave>1.2m/s, E/IVRT>2.5, E/E'>11.7.    
Answer to the reviewer: 
We would like to express our gratitude to the reviewer for providing valuable advice that has strengthened the manuscript. We have added more echocardiographic parameters with indirect signs of congestive heart failure in Table 3, such as PV/PA, E/A, E wave velocity, and E/IVRT in the revised manuscript to clarify MMVD staging as suggested by the reviewer.

Line 207 AP, what does it means?
Answer to the reviewer: 
We apologize for the lack of clarity. AP stands for Anterior-Posterior leaflet diameter, which we have edited in the revised manuscript. (Yellow highlight) We appreciate the reviewer’s understanding.

Line 216 LA decreased indicating improvement. Line 230 left ventricle dimension increase indicating clinical improvement. Could you explaine this more acurately?
Answer to the reviewer: 
We would like to express our gratitude to the reviewer for providing valuable advice that has strengthened the manuscript. Compared to the baseline, Table 3 shows a decrease in LVIDDN, LVIDSN, LA, and LA/AO five to six months after the V-clamp operation. Moreover, the %FS increased compared to the baseline. Suggesting that a V-clamp operation could improve LV dimension by decreasing preload, resulting in increased cardiac contraction, using the Frank Staring Law theory. 

Line 217 A canine patient in Stage C remains in Stage C although they do not have congestive heart failure signs. However, this might need a redefinition with the developing of this new mitral valve repairing techniques. 
Answer to the reviewer: 
We apologize for the lack of clarity on this point. We've added this point as suggested by the reviewer.

Line 221 as line 204.
Answer to the reviewer: 
We would like to express our gratitude to the reviewer for providing valuable advice that has strengthened the manuscript. We have added more echocardiographic parameters in revised manuscript.

Line 224 as line 207.
Answer to the reviewer: 
We would like to express our gratitude to the reviewer for providing valuable advice that has strengthened the manuscript. We have added more echocardiographic parameters in revised manuscript.

In line 332 you mentioned the small number of patients, this precludes a statistical analysis of results, it should be mention. 
Answer to the reviewer: 
We appreciate the thoughtful comment and apologize for the lack of clarity. We have edited the statistical analysis in the materials and method section of the revised manuscript. (Yellow highlight) We appreciate the reviewer’s understanding.
Statistical Analysis
Data were represented as mean ± standard deviation (SD). The X column represents time, and the response at each time point is entered in 4 patients for the Echocardiography parameters using GraphPad Prism 9 software.

Discussion section needs rewritting. In this section authors must comment theirs results compared to similar results from other studies, and if there are not other studies to compare with, which is the case, address the relevance of their results according to the features assess, decrease/increase in heart chamber remodeling and why, side effects of the procedure observed, why they can happen and how to approach or avoid them... Nothing is mentioned on the use of antithrombotic therapy on this paper patients but it is part of the discussion. 
Answer to the reviewer: 
We appreciate the thoughtful comment and apologize for the lack of clarity. We have edited the discussion section of the revised manuscript. (Yellow highlight) We appreciate the reviewer’s understanding.
The conclusion can be just sharing one experience. According to your manuscript only one patient suffer an arrythmic episode and pain. The conclusion refers to the objectives of the study, but they are not clarify on the text. I figure they would be the safety of the procedure, outcome of the patients, reversing of cardiac remodeling and few side effects. 
Answer to the reviewer: 
We appreciate the thoughtful comment and apologize for the lack of clarity. We have edited the revised manuscript. (Yellow highlight) We appreciate the reviewer’s understanding.
In this study, we share our perioperative experience, in managing patients before and after surgery as well as managing complications such as pain and arrhythmias. Postoperative pain management including thrombus prevention is important in improving patient survival after surgery.

One of the references used by the authors could be used as a guide as to how to write and present data on a manuscript, reference number 12. This paper is similar to the study present here but with healthy dogs.
Sasaki, K., Ma, D., Mandour, A. S., Ozai, Y., Yoshida, T., Matsuura, K., Takeuchi, A., Cheng, C. J., El-Husseiny, 396 H. M., Hendawy, H., Shimada, K., Hamabe, L., Uemura, A., & Tanaka, R. (2021). Evaluation of Changes in the 397 Cardiac Function before and after Transcatheter Edge-to-Edge Mitral Valve Repair in Healthy Dogs: Conven-398 tional and Novel Echocardiography. Animals : an open access journal from MDPI, 12(1), 56. 399 https://doi.org/10.3390/ani12010056.
Answer to the reviewer: 
We appreciate the thoughtful comment and apologize for the lack of clarity. We have edited the reference in the revised manuscript. (Yellow highlight) We appreciate the reviewer’s understanding.

Comments on the Quality of English Language
Overall, language is fine but some paragraphs needs rewriting, use of appropiate and correct positioning of words.
We would like to thank the reviewer for this helpful advice to strengthen the manuscript. We’ve checked the English as the logical structure of the sentences thoroughly in the revised manuscript as suggested by the reviewer.  However, we have asked an English native speaker who has experience in writing manuscripts in biomedical science to extensively proofread the revised manuscript as suggested by the reviewer.  We believe that the revised manuscript has been much improved regarding the English writing.

Best Regards
Soontaree Petchdee
Corresponding Author

Reviewer 3 Report

Comments and Suggestions for Authors

Thank you for this submission.  The more we can understand about the outcomes of TEER in dogs, the better.  The two main areas lacking in this report are some of the specifics related to the surgical procedure, for example how surgeons determined the appropriate puncture site at the apex, and how how the puncture was performed (was a purse-string placed in the apex, were pledgets used, etc).  Also, there is no information on post-operative use of cardiac medications.  Was pimobendan continued after surgery, and if so, for how long?  Were all dogs started on clopidogrel, and for how long?  Understanding the long-term post-operative management is very important for procedural success.  I have attached a PDF with comments and questions to be considered.  

Comments on the Quality of English Language

In general the paper reads well, however there are a few sentences that could use revision, as well as some spelling and grammatical errors that need attention. 

Author Response

Dear Editor,
We have provided a point-by-point reply to each point raised by the reviewers as a separate file and indicated any changes or corrections within the text, highlighting the resubmitted manuscript in yellow. Please see below for the list of changes. 
Reviewer 3
Comments and Suggestions for Authors
Thank you for this submission.  The more we can understand about the outcomes of TEER in dogs, the better.  The two main areas lacking in this report are some of the specifics related to the surgical procedure, for example how surgeons determined the appropriate puncture site at the apex, and how how the puncture was performed (was a purse-string placed in the apex, were pledgets used, etc).  Also, there is no information on post-operative use of cardiac medications.  Was pimobendan continued after surgery, and if so, for how long?  Were all dogs started on clopidogrel, and for how long?  Understanding the long-term post-operative management is very important for procedural success.  I have attached a PDF with comments and questions to be considered.  
We would like to thank the reviewer for his/her positive comment on the significance of our findings and would like to thank the reviewer for this helpful advice to strengthen the manuscript.  We’ve revised and checked that all comments have been clarified in the revised manuscript as suggested by the reviewer. Below are our responses to the reviewer’s suggestions
This sentence does not make sense.
this may not be a universal recommendation, so consider modifying to say that "some recommend surgery for patients with B2 disease"
as well as costly to owners without pet insurance.
We apologize for the lack of clarity on this point. We've revised this point as suggested by the reviewer.
The Surgical intervention
can more detail be provided regarding the surgical approach?  For example, decision-making for intercostal incision site, size of incision, how to determine where to make apical puncture...
We apologize for the lack of clarity on this point. We've revised this point as suggested by the reviewer.
Has this questionnaire been used in dogs previously?  Also, this is introduced here, but there is no discussion of the scores provided in the table.
We apologize for the lack of clarity on this point. We've revised this point in discussion part as suggested by the reviewer.

Table 1: This table is confusing to read, and does not really show improvement in any scores for before and after.  It would be helpful to include a discussion of the results of this table in the discussion.
We apologize for the lack of clarity on this point. We've revised this point in discussion part as suggested by the reviewer.
Table 3: I think it would be more impactful to share the actual measurements at 4 months rather than % change, or providing the 4 month measurement with % change in parentheses so the table size is not significantly impacted.
We apologize for the lack of clarity on this point. We've revised this point as suggested by the reviewer.
Can you provide any information on medications the patients were on at discharge?
We apologize for the lack of clarity on this point. We've revised this point as suggested by the reviewer.
Is this a typo?
We apologize for the typo mistake. We've corrected as suggested by the reviewer.
were these patients considered high risk?  In the methods you describe assessing general health, but you do not comment on the findings of this general heath assessment in the results.  If not, you cannot state that this procedure is safe for high risk patients based on this series.
We apologize for the lack of clarity on this point. We've revised this point as suggested by the reviewer.
should have brackets
We apologize for the typo mistake. We've corrected as suggested by the reviewer.
Not sure this is a unique technique or strategy such that it needs images in the manuscript.  It would be more interesting to include a figure showing the left apical puncture or something unique associated with the v-clamp procedure.
We apologize for the lack of clarity on this point. We've revised figure 3 as suggested by the reviewer.
This is not true for most interventions in veterinary medicine.  Routine procedures such as balloon valvuloplasty, ACDO for PDA for example do not require post-operative anticoagulation
We apologize for the lack of clarity on this point. We've revised this point as suggested by the reviewer.
were any other anticoagulants used, or just clopidogrel?
We apologize for the lack of clarity on this point. We've revised this point as suggested by the reviewer.
Comments on the Quality of English Language
In general, the paper reads well, however, there are a few sentences that could use revision, as well as some spelling and grammatical errors that need attention.
Answer to the reviewer: 
We would like to thank the reviewer for this helpful advice to strengthen the manuscript. We’ve checked the English as the logical structure of the sentences thoroughly in the revised manuscript as suggested by the reviewer.  However, we have asked an English native speaker who has experience in writing manuscripts in biomedical science to extensively proofread the revised manuscript as suggested by the reviewer.  We believe that the revised manuscript has been much improved regarding the English writing.

Best Regards
Soontaree Petchdee
Corresponding Author

Reviewer 4 Report

Comments and Suggestions for Authors

The author use a transcatheter edge to edge repair device to mimic the open heart mitral valveplasty surgery in the canine. This research might be the very first transcatheter device implanted in Thailand. The technique immediately repairs the mitral valve and can be used as another treatment option to reduce the incidence of complications of open-heart surgery in dogs, the results suggest that the procedure is safe, potentially eliminating the need for heart failure medication after repair.  The innovative part of this research is that the author uses the questionnaire to help the owner evaluate the QOL of our canine patients. These results had never been reported before. Previous research mainly concentrates on cardiac function and clinical signs, which have also been reviewed in this research. This might raise the interest of potential animal owners in this procedure.  The E/IVRT value used in this research is a great tool for assessing LAP. Mainly, this research proved the utility and safety of transcatheter edge-to-edge repair devices in canines. 

Comments on the Quality of English Language

Only minor mistakes were recognized and it did not affect reading.

Author Response

We thank the reviewers for their feedback and this useful advice.

Reviewer 5 Report

Comments and Suggestions for Authors

In the work titled “Transcatheter edge-to-edge repair of the mitral valve in four dogs”, the authors reported 4 interesting cases of dogs with severe mitral regurgitation, who were symptomatic for heart failure and underwent transcatheter edge-to-edge mitral valve repair with the ValveClamp device. For each case, detailed information on dogs’ pre-procedural clinical status, pre-procedural echocardiographic evaluation, interventions and follow-up were analyzed and reported.

The paper is well structured, and the discussion section contains adequate references to current evidence.

My only concern is related to the quality of English language. A careful editing of English language is required.

Comments on the Quality of English Language

My only concern is related to the quality of English language. A careful editing of English language is required.

Author Response

We would like to thank the reviewer for this helpful advice to strengthen the manuscript. We’ve checked the English as the logical structure of the sentences thoroughly in the revised manuscript as suggested by the reviewer.  However, we have asked an English native speaker who has experience in writing manuscripts in biomedical science to extensively proofread the revised manuscript as suggested by the reviewer.  We believe that the revised manuscript has been much improved regarding the English writing

Round 2

Reviewer 1 Report

Comments and Suggestions for Authors

The authors focus on the clinical course before and after surgery, but descriptions and discussions of perioperative arterial lines remain. The description of the method contains the reasons for the discussion and methodology, and the description of the results is scattered in the discussion section, and it does not form the form of a thesis. For the above reasons, I judge that it is not accepted because the revision that leads to acceptance is not allowed.

Comments on the Quality of English Language

I judge that it is not accepted because the revision that leads to acceptance is not allowed.

Author Response

We thank the reviewers for their feedback and this useful advice. We hope that the revised manuscript has been much improved regarding the English writing.

Reviewer 2 Report

Comments and Suggestions for Authors

I would like to thank the authors for submitting a review of their manuscripts. The manuscript has improved considerably but, I would like to suggest further changes to make it more appealing, and easy to read and understand.

Line 11-12 Medical treatment is the initial therapy, and it is also the therapy for congestive heart failure. I suggest a change here that could be as follows: ‘Mitral valve regurgitation is a common prevalent heart condition in dogs that often progresses to heart failure. Although medical treatment is the recommended therapy, in the past few years…’

Lines 22-23 underwent surgical mitral valve repair between December 2023 and March 2024.

Line 26 Echocardiography was conducted at baseline and on the 6 month that follow the procedure.

Line 43-44 This sentence does not make sense. Needs rewriting.

Line 45 is invasive, time-consuming and, costly…

Line 64 there is no reference to Minnesotans Living with heart failure and validation in the Introduction, as indicated on the first review. The use of questionnaires is common in Human Medicine and it is becoming so in Veterinary Medicine. An introduction to this type of questionnaires and why they are used should be given to the readers, as authors consider it an essential part of their work.

After Line 64 To wrap up the Introduction section I would recommend the authors to say why this manuscript, what is their goal, the objectives of this paper. Is it to try to demonstrate that the V-clamp is feasible and safe, is to to say that heart enlargement involves after V-Clamp placement,  is it to say that after V-clamp dogs live better according to Minnesotans Living… questionnaire…?

Line 68 I would recommend authors to start this section with lines 76-77 and then I would recommend the authors to continue with “Evaluation” lines 178-183. And to use the correct linking words.

Line 141 Butorphanol 0,2mg/kg (Zoetis, Xmg/ml)…etomidate 1-2mg/kg (B Braun Xmg/ml)…bupivacaine at ribs 5, 6, 7, 8, and 9 (Aspen Xmg/ml)…

Line 145-146 would be part of a discussion and not the methods and material.

Line 161 For 24 hours after surgery? Incorrect writing. Suggestion: Twenty four hours after surgery…

Line 173-176 I could not find any X column on the Tables presented. Only on one occasion mean and standard deviation is provided, and this should not be considered a statistical analysis.

Line 182 Figure 2. This has not been changed, Figure 2 is V-Clamp placement.

Lines 185 This line introduces the use of questionnaire. As mentioned, there should be a reference to them in the introduction.

Line 192 is part of results.

Lines 194-197. I would recommend this information to go in the results section after the patients’ outcome information. 

Table 2. Add AP meaning to the Table caption.

Line 274-277 repeats the previous lines.

Figure 4 If the results presented on Figure 4 refers to a particular patient it should be mentioned. It Figure 4 refers to a median and SD deviation value of the four dogs it should be mentioned as well and, represent  the median and SD deviation value of the four dogs having been subjected to the same number of follow-ups.

Lines 282-283. Repeats the information given on the presentation of the patients, on the Material and Methods section. Not needed.

Lines 288-289. Same as before.

Lines 296-297 Same as before.

Lines 303-304 Same as before.

Lines 313 time is shortened… Should it not be ‘was’?

Line 314 ratio is reduced… Should it not be ‘was’?

Line 334. No much difference on Hygiene and physical capacity for Stage B2 dogs either. With the data provided on Table 1, subjectively.

Line 336-337. What do you try to say here?

Line 337-338. I disagree with that assumption. The data provided do not show that decline. What do you try to say with the sentence ‘and should be considered for further surgical treatment’, that V-clamp procedure makes a dog unhappy, depressed…?

Line 345 In this study… Is it really needs on the sentence?

Line 379 anesthetic plan instead of anesthesia plan.

Line 383-384 and Figure 5 do not add relevant information. Those lines and Figure 5 could be omitted to reduce the extension of the manuscript.

Line 391-393 The authors try to give too much information and the result is not understandable. They could divide the paragraph in two smaller sentences that make sense.

Line 406 If authors are suggesting something it would be advisable to use the form could instead of can.

Line 426 interpreted the results. Interpreted the results…

 A different order of paragraphs would give cohesion and sense to the information provided, and would make the text easy to read. 

Comments on the Quality of English Language

Regarding English language I recommend the following:

Authors should try to express their ideas individually, a sentence for each point that needs mention.

Use linking words when they are needed and omit them when a punctuation sign can be used instead. 

Author Response

Reviewer 2

I would like to thank the authors for submitting a review of their manuscripts. The manuscript has improved considerably but, I would like to suggest further changes to make it more appealing, and easy to read and understand.

Line 11-12 Medical treatment is the initial therapy, and it is also the therapy for congestive heart failure. I suggest a change here that could be as follows: ‘Mitral valve regurgitation is a common prevalent heart condition in dogs that often progresses to heart failure. Although medical treatment is the recommended therapy, in the past few years…’

Answer to the reviewer:

We would like to thank the reviewer for his/her positive comment and would like to thank the reviewer for this helpful advice to strengthen the manuscript.  We’ve revised the revised manuscript as suggested by the reviewer.

Lines 22-23 underwent surgical mitral valve repair between December 2023 and March 2024.

Answer to the reviewer:

We would like to thank the reviewer for his/her positive comment and would like to thank the reviewer for this helpful advice to strengthen the manuscript.  We’ve revised the revised manuscript as suggested by the reviewer.

Line 26 Echocardiography was conducted at baseline and on the 6 months that follow the procedure.

Answer to the reviewer:

We would like to thank the reviewer for his/her positive comment and would like to thank the reviewer for this helpful advice to strengthen the manuscript.  We’ve revised the revised manuscript as suggested by the reviewer.

Line 43-44 This sentence does not make sense. Needs rewriting.

Answer to the reviewer:

We would like to thank the reviewer for his/her positive comment and would like to thank the reviewer for this helpful advice to strengthen the manuscript.  We’ve revised the revised manuscript as suggested by the reviewer.

Line 45 is invasive, time-consuming and, costly…

Answer to the reviewer:

We would like to thank the reviewer for his/her positive comment and would like to thank the reviewer for this helpful advice to strengthen the manuscript.  We’ve revised the revised manuscript as suggested by the reviewer.

Line 64 there is no reference to Minnesotans Living with heart failure and validation in the Introduction, as indicated on the first review. The use of questionnaires is common in Human Medicine and it is becoming so in Veterinary Medicine. An introduction to this type of questionnaires and why they are used should be given to the readers, as authors consider it an essential part of their work.

Answer to the reviewer:

We would like to thank the reviewer for his/her positive comment and would like to thank the reviewer for this helpful advice to strengthen the manuscript.  We’ve revised the revised manuscript as suggested by the reviewer.

The Minnesotans Living with Heart Failure and Validation has been extensively used, demon-strating strong reliability and validity in assessing health-related quality of life in chronic heart failure patients in humans [9]. The Minnesotans Living with Heart Failure and Validation are scored by adding the individual item scores for each dimension, resulting in a total score as well as separate scores for physical and emotional well-being. A higher total score indicates a lower overall quality of life. The evaluated results from the questionnaires which were adapted from Minnesotans Living with Heart Failure and Validation may help for further consideration of postoperative quality of life as an important goal and can be measured as an adjunct to quantitative outcomes for dogs.

After Line 64 To wrap up the Introduction section I would recommend the authors to say why this manuscript, what is their goal, the objectives of this paper. Is it to try to demonstrate that the V-clamp is feasible and safe, is to to say that heart enlargement involves after V-Clamp placement,  is it to say that after V-clamp dogs live better according to Minnesotans Living… questionnaire…?

Answer to the reviewer:

We would like to thank the reviewer for his/her positive comment and would like to thank the reviewer for this helpful advice to strengthen the manuscript.  We’ve revised the revised manuscript as suggested by the reviewer.

Line 68 I would recommend authors to start this section with lines 76-77 and then I would recommend the authors to continue with “Evaluation” lines 178-183. And to use the correct linking words.

Answer to the reviewer:

We would like to thank the reviewer for his/her positive comment and would like to thank the reviewer for this helpful advice to strengthen the manuscript.  We’ve revised the revised manuscript as suggested by the reviewer.

Line 141 Butorphanol 0,2mg/kg (Zoetis, Xmg/ml)…etomidate 1-2mg/kg (B Braun Xmg/ml)…bupivacaine at ribs 5, 6, 7, 8, and 9 (Aspen Xmg/ml)…

Answer to the reviewer:

We would like to thank the reviewer for his/her positive comment and would like to thank the reviewer for this helpful advice to strengthen the manuscript.  We’ve revised the revised manuscript as suggested by the reviewer.

Line 145-146 would be part of a discussion and not the methods and material.

Answer to the reviewer:

We would like to thank the reviewer for his/her positive comment and would like to thank the reviewer for this helpful advice to strengthen the manuscript.  We’ve revised the revised manuscript as suggested by the reviewer.

Line 161 For 24 hours after surgery? Incorrect writing. Suggestion: Twenty four hours after surgery…

Answer to the reviewer:

We would like to thank the reviewer for his/her positive comment and would like to thank the reviewer for this helpful advice to strengthen the manuscript.  We’ve revised the revised manuscript as suggested by the reviewer.

Line 173-176 I could not find any X column on the Tables presented. Only on one occasion mean and standard deviation is provided, and this should not be considered a statistical analysis.

Answer to the reviewer:

We apologize for the lack of clarity on this point. We've added this point in figure 4 as suggested by the reviewer.

Line 182 Figure 2. This has not been changed, Figure 2 is V-Clamp placement.

We apologize for the mistake on this point. We've added this point in figure 2 as suggested by the reviewer.

Lines 185 This line introduces the use of questionnaire. As mentioned, there should be a reference to them in the introduction.

Answer to the reviewer:

We would like to thank the reviewer for his/her positive comment and would like to thank the reviewer for this helpful advice to strengthen the manuscript.  We’ve revised the revised manuscript as suggested by the reviewer.

Line 192 is part of results.

Lines 194-197. I would recommend this information to go in the results section after the patients’ outcome information.

Answer to the reviewer:

We would like to thank the reviewer for his/her positive comment and would like to thank the reviewer for this helpful advice to strengthen the manuscript.  We’ve revised the revised manuscript as suggested by the reviewer by moving to the results section.

Table 2. Add AP meaning to the Table caption.

Answer to the reviewer:

We would like to thank the reviewer for his/her positive comment and would like to thank the reviewer for this helpful advice to strengthen the manuscript.  We’ve revised the revised manuscript as suggested by the reviewer.

Line 274-277 repeats the previous lines.

Answer to the reviewer:

We would like to thank the reviewer for this helpful advice to strengthen the manuscript.  We’ve revised the revised manuscript as suggested by the reviewer.

Figure 4 If the results presented on Figure 4 refers to a particular patient it should be mentioned. It Figure 4 refers to a median and SD deviation value of the four dogs it should be mentioned as well and, represent the median and SD deviation value of the four dogs having been subjected to the same number of follow-ups.

Lines 282-283. Repeats the information given on the presentation of the patients, on the Material and Methods section. Not needed. Lines 288-289. Same as before. Lines 296-297 Same as before. Lines 303-304 Same as before.

Answer to the reviewer:

We would like to thank the reviewer for his/her positive comment and would like to thank the reviewer for this helpful advice to strengthen the manuscript.  We’ve revised the revised manuscript as suggested by the reviewer.

Lines 313 time is shortened… Should it not be ‘was’? Line 314 ratio is reduced… Should it not be ‘was’?

Answer to the reviewer:

We would like to thank the reviewer for his/her positive comment and would like to thank the reviewer for this helpful advice to strengthen the manuscript.  We’ve revised the revised manuscript as suggested by the reviewer.

Line 334. No much difference on Hygiene and physical capacity for Stage B2 dogs either. With the data provided on Table 1, subjectively.

Answer to the reviewer:

We would like to thank the reviewer for his/her positive comment and would like to thank the reviewer for this helpful advice to strengthen the manuscript.  We’ve revised the revised manuscript as suggested by the reviewer.

Line 336-337. What do you try to say here?

Answer to the reviewer:

We apologize for the lack of clarity on this point. We edited this point in the revised manuscript as reviewer’s comments. We appreciate the reviewer’s understanding.

Line 337-338. I disagree with that assumption. The data provided do not show that decline. What do you try to say with the sentence ‘and should be considered for further surgical treatment’, that V-clamp procedure makes a dog unhappy, depressed…?

Answer to the reviewer:

We apologize for the lack of clarity on this point. We edited this point in the revised manuscript as reviewer’s comments. We appreciate the reviewer’s understanding.

Line 345 In this study… Is it really needs on the sentence?

Answer to the reviewer:

We apologize for the lack of clarity on this point. We removed the sentences in the revised manuscript as reviewer’s comments. We appreciate the reviewer’s understanding.

Line 379 anesthetic plan instead of anesthesia plan.

Answer to the reviewer:

We apologize for the mistake. We edited the method section in the revised manuscript as

reviewer’s comments. We appreciate the reviewer’s understanding.

Line 383-384 and Figure 5 do not add relevant information. Those lines and Figure 5 could be omitted to reduce the extension of the manuscript.

We would like to thank the reviewer for his/her positive comment and would like to thank the reviewer for this helpful advice to strengthen the manuscript.  We’ve deleted Figure 5 in the revised manuscript as suggested by the reviewer.

Line 391-393 The authors try to give too much information and the result is not understandable. They could divide the paragraph in two smaller sentences that make sense.

Answer to the reviewer:

We would like to thank the reviewer for his/her positive comment and would like to thank the reviewer for this helpful advice to strengthen the manuscript.  We’ve revised the revised manuscript as suggested by the reviewer.

Line 406 If authors are suggesting something it would be advisable to use the form could instead of can.

Answer to the reviewer:

We would like to thank the reviewer for his/her positive comment and would like to thank the reviewer for this helpful advice to strengthen the manuscript.  We’ve revised the revised manuscript as suggested by the reviewer.

Line 426 interpreted the results. Interpreted the results…

We apologize for the mistake. We edited the method section in the revised manuscript as

reviewer’s comments. We appreciate the reviewer’s understanding.

A different order of paragraphs would give cohesion and sense to the information provided, and would make the text easy to read.

Answer to the reviewer:

We would like to thank the reviewer for his/her positive comment and would like to thank the reviewer for this helpful advice to strengthen the manuscript.  We’ve revised the revised manuscript as suggested by the reviewer.

Comments on the Quality of English Language

Regarding English language I recommend the following:

Authors should try to express their ideas individually, a sentence for each point that needs mention.

Use linking words when they are needed and omit them when a punctuation sign can be used instead.

Answer to the reviewer:

We would like to thank the reviewer for this helpful advice to strengthen the manuscript. We’ve checked the English as the logical structure of the sentences thoroughly in the revised manuscript as suggested by the reviewer.  However, we have asked an English native speaker who has experience in writing manuscripts in biomedical science to extensively proofread the revised manuscript as suggested by the reviewer.  We believe that the revised manuscript has been much improved regarding the English writing.

Best Regards

Soontaree Petchdee

Reviewer 3 Report

Comments and Suggestions for Authors

Thank you to the authors for their continued efforts to improve the manuscript.  In the current state, some improvements could be made to the organization of the information.  For example, the entire heading "Evaluation" can be eliminated and the contents can be divided between methods and results.  Lines 177-192 would fit more appropriately in the methods section, and the remainder could be moved to results.  

Similar to the previous review, I find the inclusion of the images detailing placement of an arterial line unnecessary.  It is not a particularly novel and there is no discussion of the use of arterial blood pressure in the methods section, so it is not clear what the point of including the images is.  

Tables 2 and 3 need adjustments to formatting, it appears that new information added affected the columns such that values do not line up with appropriate headings.  

I also question the inclusion of QOL survey since this survey has not been validated in animals.  The results are confusing at best and don't really add anything to the results of the paper.  I think that reporting the changes in echo parameters and ability to discontinue diuretic and pimobendan are strong enough data to support the continued use and investigation of TEER in dogs.  

I have included a PDF with comments for additional consideration.

Comments on the Quality of English Language

Some minor improvements to the English grammar could still be made.

Author Response

Reviewer 3

Thank you to the authors for their continued efforts to improve the manuscript.  In the current state, some improvements could be made to the organization of the information.  For example, the entire heading "Evaluation" can be eliminated and the contents can be divided between methods and results.  Lines 177-192 would fit more appropriately in the methods section, and the remainder could be moved to results. 

We apologize for the inappropriate heading. We've revised this point as suggested by the reviewer.

Similar to the previous review, I find the inclusion of the images detailing placement of an arterial line unnecessary.  It is not particularly novel and there is no discussion of the use of arterial blood pressure in the methods section, so it is not clear what the point of including the images is. 

We apologize for the unnecessary figure. We would like to thank the reviewer for this helpful advice to strengthen the manuscript.

Tables 2 and 3 need adjustments to formatting, it appears that new information added affected the columns such that values do not line up with appropriate headings. 

We apologize for the inappropriate heading. We've revised this point as suggested by the reviewer.

I also question the inclusion of QOL survey since this survey has not been validated in animals.  The results are confusing at best and don't really add anything to the results of the paper.  I think that reporting the changes in echo parameters and ability to discontinue diuretic and pimobendan are strong enough data to support the continued use and investigation of TEER in dogs. 

We apologize for the unnecessary results. We would like to thank the reviewer for this helpful advice to strengthen the manuscript.

Best Regards

Soontaree Petchdee
